# Emergent Symbolic Mechanisms Support Abstract Reasoning in Large Language Models

**Yukang Yang** [1] **Declan Campbell** [2] **Kaixuan Huang** [1] **Mengdi Wang** [1] **Jonathan Cohen** [2] **Taylor Webb** [3]

## Abstract

Many recent studies have found evidence for emergent reasoning capabilities in large language models (LLMs), but debate persists concerning the robustness of these capabilities, and the extent to which they depend on structured reasoning mechanisms. To shed light on these issues, we study the internal mechanisms that support abstract reasoning in LLMs. We identify an emergent symbolic architecture that implements abstract reasoning via a series of three computations. In early layers, *symbol abstraction heads* convert input tokens to abstract variables based on the relations between those tokens. In intermediate layers, *symbolic induction heads* perform sequence induction over these abstract variables. Finally, in later layers, *retrieval heads* predict the next token by retrieving the value associated with the predicted abstract variable. These results point toward a resolution of the longstanding debate between symbolic and neural network approaches, suggesting that emergent reasoning in neural networks depends on the emergence of symbolic mechanisms.

## 1. Introduction

Large language models (LLMs) have become the dominant paradigm in artificial intelligence, but there is ongoing debate concerning the limits and reliability of their capabilities. One major focus of this debate has been the question of whether they can reason systematically in an abstract or human-like manner. Many studies have documented impressive performance on various reasoning tasks (Wei et al.,

[1]Department of Electrical and Computer Engineering, Princeton University, Princeton, NJ [2]Princeton Neuroscience Institute, Princeton University, Princeton, NJ [3]Microsoft Research, New York, NY. Correspondence to: Taylor Webb <taylor.w.webb@gmail.com>.

*Proceedings of the 42$^{nd}$ International Conference on Machine Learning*, Vancouver, Canada. PMLR 267, 2025. Copyright 2025 by the author(s).

2022; Mirchandani et al., 2023), even rivaling human performance in some cases (Webb et al., 2023; Musker et al., 2024; Webb et al., 2025), but other studies have questioned these conclusions (Wu et al., 2024; McCoy et al., 2023; Lewis & Mitchell, 2024). In particular, LLMs appear to perform more poorly in some reasoning domains, such as mathematical reasoning (Dziri et al., 2024) or planning (Momennejad et al., 2024); and, even in domains in which they have shown strong performance such as analogical reasoning (Webb et al., 2023), some studies have questioned the robustness of these capabilities (Lewis & Mitchell, 2024).

These conflicting findings raise the question: are LLMs genuinely capable of structured human-like reasoning, or are they merely mimicking this capacity by statistically approximating their training data? One way to answer this question is to look at the internal mechanisms that support this capacity. It has long been hypothesized that innate symbol-processing mechanisms are required to support human-like abstraction (Marcus, 2001; Dehaene et al., 2022; Wong et al., 2023). It has also been demonstrated that neural networks are capable, at least in principle, of implementing some of the key properties of symbolic systems (Smolensky, 1990; Hummel & Holyoak, 2003; Kriete et al., 2013), and that the incorporation of these properties as architectural inductive biases can support data-efficient acquisition of abstract symbolic reasoning (Webb et al., 2021; Altabaa et al., 2024; Webb et al., 2024). It remains unclear, however, in the case of transformer-based LLMs that do not obviously possess such strong inductive biases, what mechanisms support their emergent capability for abstraction.

Here, we report evidence for a set of emergent symbolic mechanisms that support abstract reasoning in LLMs. Specifically, we identify an emergent three-stage architecture consisting of the following mechanisms: 1) in early layers, *symbol abstraction heads* convert input tokens to abstract variables (i.e., symbols) based on the relations between those tokens, 2) in intermediate layers, *symbolic induction heads*, perform sequence induction over these variables, and 3) in later layers, *retrieval heads* perform next-token prediction by retrieving the value associated with the predicted variable. These mechanisms capture two key properties of symbol processing. First, they operate over

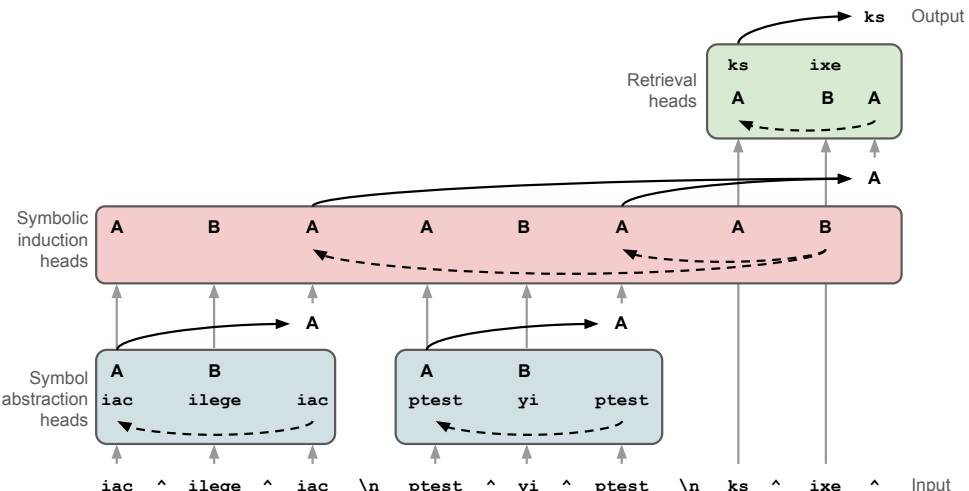

*Figure 1.* **Emergent Symbolic Architecture**. Schematic depiction of the proposed three-stage architecture for abstract reasoning in language models. Example depicts architecture as applied to an algebraic rule induction task, involving either ABA or ABB rules instantiated with random tokens. Symbol abstraction heads identify relations between input tokens and, based on these relations, represent the tokens using a consistent set of abstract variables aligned with their role in the relations. Symbolic induction heads perform sequence induction over abstract variables (i.e., they predict the next variable based on the sequence observed in the previous in-context examples). Retrieval heads predict the next token by retrieving the value associated with the predicted abstract variable.

representations of abstract variables that are *invariant* to the values that they are associated with. That is, the representation of an abstract variable is the same regardless of which values instantiate that variable. Second, these mechanisms utilize *indirection*, meaning that variables refer to content that is stored at a different location than the variables themselves (i.e., they are pointers).

To identify and validate the presence of these mechanisms, we perform experiments across three abstract reasoning tasks – algebraic rule induction, letter string analogies, and verbal analogies – and 13 open-source LLMs from four model families – GPT-2 (Radford et al., 2019), Gemma-2 (Gemma Team, 2024), Qwen2.5 (Qwen Team, 2025), and Llama-3.1 (Dubey et al., 2024) – drawing on convergent evidence from a series of causal, representational, and attention analyses. We find robust evidence for these mechanisms across all three tasks, and three out of four model families (Gemma-2, Qwen2.5, and Llama-3.1; with more equivocal results for GPT-2). Taken together, these results suggest that emergent reasoning in LLMs depends on structured, abstract reasoning mechanisms, rather than simple statistical approximation. More broadly, these results suggest a resolution to the longstanding debate between symbolic and neural network approaches, illustrating how neural networks can learn to perform abstract reasoning via the development of emergent symbol processing mechanisms.

## 2. Approach

Figure 1 depicts the proposed architecture as applied to an algebraic rule induction task involving sequences governed by one of two identity rules, ABA or ABB. For each problem, two in-context examples were presented, followed by an incomplete third example. The model was expected to generate the token that completes this third example. We instantiated rules using tokens randomly sampled from each model's vocabulary (ensuring that in-context examples within the same problem instance did not share tokens). We begin by presenting results from this rule induction task and the Llama-3.1 70B model, and then present results for additional tasks and models.

We found that Llama-3.1 70B displayed a 2-shot accuracy of 95% on the rule induction task. Although this task is relatively simple, especially when compared with some of the tasks that have been featured in recent debates over LLM reasoning (e.g., matrix reasoning (Webb et al., 2023)), it nevertheless offers a paradigmatic case of relational abstraction. In particular, the use of completely arbitrary tokens ensures that the task cannot be solved by relying on statistical patterns specific to the tokens or associations among them, and for this reason it has previously been used to argue for the presence of symbol-processing mechanisms in human cognition (Marcus et al., 1999), and to evaluate systematic generalization of abstract rules in neural networks (Webb et al., 2021). Accordingly, the ability to reliably solve this

task is already strongly suggestive of the presence of some form of symbol-processing. In the following sections, we describe a specific mechanistic hypothesis for how symbol-processing might be carried out in this model.

## 2.1. Symbol Abstraction Heads

Our hypothesis consists of three stages. In the first stage, input tokens are converted to symbolic representations. The inspiration for this hypothesis comes from the *abstractor* architecture (Altabaa et al., 2024), a variant of the transformer that implements a strong relational inductive bias (Webb et al., 2024). In that architecture, a modified form of attention (termed *relational cross-attention*) is employed in which the values consist of a standalone set of learned embeddings, rather than being conditioned on the input tokens as in standard self-attention. As a result, the output of this attention operation is completely abstracted away from the identity of the input tokens, and instead only reflects the pattern of relations among those tokens (as encoded by the pattern of inner products between query and key embeddings). These outputs can therefore be viewed as a form of learned, distributed symbolic representations.

Here, we hypothesize that an emergent form of this relational attention operation is implemented by attention heads in early layers of the model. We refer to these heads as *symbol abstraction heads*. Concretely, the keys and query embeddings in these heads represent the input tokens, and the inner product between keys and queries represents the relations between these tokens. It is natural to interpret this operation as representing similarity relations (and this is the relevant type of relations in the rule induction task), but it is also possible for this operation to represent a broader class of relations (Altabaa & Lafferty, 2024) (as necessitated by some of the other tasks that we investigate). Importantly, we hypothesize that the value embeddings in these heads do *not* carry information about the specific identity of the input tokens (i.e., they are *invariant* to the content of those tokens), but instead represent only their position. More precisely, we hypothesize that the value embeddings represent the relative position of a token within an in-context example, as this is precisely the information that's needed to compute the abstract variable associated with that token (e.g., the fact that the first token and the third token are the same in an ABA rule is precisely what determines that they share the same variable). Given that these conditions are met, the self-attention operation is equivalent to relational cross-attention (Altabaa et al., 2024), and the output of such an attention head will represent an abstract variable.

## 2.2. Symbolic Induction Heads

In the second stage, we hypothesize that sequence induction is performed over the abstract variables computed in the first stage. This hypothesis is inspired by previous work on *induction heads*, an emergent circuit that supports in-context learning in transformers (Elhage et al., 2021; Olsson et al., 2022). As originally formulated, this circuit performs a simple sequence induction mechanism: given a sequence that ends with a particular token, an induction head will look for previous instances of that token, and retrieve the token that succeeded it. Although this mechanism performs induction based only on in-context bigram statistics, subsequent work has identified heads that also compute more complex n-gram statistics (Akyürek et al., 2024). Here, we use the term 'induction' to refer to the more general process of predicting the next token based on in-context transition probabilities (i.e., beyond bigram statistics).

We hypothesize that a symbolic variant of this mechanism is responsible for performing induction over sequences of symbols rather than literal tokens. We refer to the attention heads that carry out this mechanism as *symbolic induction heads*. Unlike standard induction heads, which operate over direct representations of the input tokens, symbolic induction heads operate over representations of abstract variables (computed by symbol abstraction heads in previous layers). The output of symbolic induction heads is a prediction of the abstract variable associated with the next token. Empirically, we find that symbolic induction heads are distinct from standard induction heads (section 3.5).

## 2.3. Retrieval Heads

Finally, in the third stage, we hypothesize that a separate mechanism is used to convert the abstract variables (symbols) to their associated tokens (values), by performing a simple form of retrieval. We refer to the attention heads that perform this retrieval operation as *retrieval heads*. The key and query embeddings in these heads represent abstract variables, and the value embeddings represent the corresponding input tokens. Retrieval heads perform the inverse of the relational attention operation performed by symbol abstraction heads. Given an input embedding representing an abstract variable (the prediction computed by symbolic induction heads in previous layers), this variable is matched with previous instances, and the associated token is retrieved. This can be viewed as a form of *indirection*, wherein a variable (i.e., a pointer to a particular location in memory) is used to retrieve the value associated with it (i.e., the data stored at that location).

## 3. Results

### 3.1. Causal Mediation Analyses

We performed causal mediation analysis (Pearl, 2022; Meng et al., 2022; Wang et al., 2023; Todd et al., 2024) to isolate the hypothesized attention heads. In this analysis, embed-

dings from one context are patched into another context. This approach can be used to estimate the causal effect of an embedding at a particular layer, position, or attention head.

Our analysis had two conditions. In one condition, intended to isolate representations of abstract variables (i.e., symbols), we created two contexts in which the same token is associated with two different variables. Given a set of tokens $A_1, B_1, ... A_N, B_N$ (where $N - 1$ represents the number of in-context examples), we created one context $c_1^{abstract}$ that instantiated an ABA rule, and another context $c_2^{abstract}$ that instantiated an ABB rule (equivalent to a BAA rule):

$$c_1^{abstract} = A_1, B_1, A_1, ..., A_N, B_N \qquad (1)$$

$$c_2^{abstract} = B_1, A_1, A_1, ..., B_N, A_N \qquad (2)$$

Importantly, in this analysis, the final token in each in-context example, $A_N$, is identical for both contexts, but it is associated with a different variable, based on its relations to the other tokens within the same in-context example.

We contrast this with another condition that is intended to isolate representations of literal tokens. Given the same set of tokens used for the previous condition, we created the following two contexts:

$$c_1^{token} = A_1, B_1, A_1, ..., A_N, B_N \qquad (3)$$

$$c_2^{token} = A_1, B_1, A_1, ..., B_N, A_N \qquad (4)$$

In this condition, both contexts involve the same abstract rule, but the tokens used in the final example are swapped. We also perform a version of both analyses using the ABB rule and average the results for the two rules.

Together, these analyses allow for a double dissociation between representations of abstract variables (abstract causal mediation) vs. the tokens associated with those variables (token causal mediation). For each analysis, given two contexts $c_1$ and $c_2$, and a pretrained lanuage model $f(\cdot)$ that outputs logits for all possible next tokens, we computed the causal mediation score developed by Wang et al. (2023):

$$s = \big(f(c_1^*)[y_{c_1^*}] - f(c_1^*)[y_{c_1}]\big)$$
$$- \big(f(c_1)[y_{c_1^*}] - f(c_1)[y_{c_1}]\big) \qquad (5)$$

where $f(c_1)[y]$ is the logit for answer $y$ in context $c_1$; $f(c_1^*)[y]$ is the logit for answer $y$ in the patched context $c_1^*$, for which activations are patched from $c_2$ to $c_1$ at specific layers, positions, and/or attention heads; $y_{c_1}$ is the correct answer for context $c_1$; and $y_{c_1^*}$ is the expected answer for the patched context $c_1^*$. Intuitively, this score reflects the extent to which patching activations from context $c_2$ to $c_1$ has the expected effects on the model's outputs. For $c^{abstract}$,

which was designed to test whether the activations represent abstract variables, we expected that activation patching would convert the *symbol* for the token that completes the incomplete query, whereas for $c^{token}$, we expected that activation patching would convert the literal *token*. See Section B.2 and Figures 5-7 for more details (including illustrated examples) on this analysis.

Figures 2(a) and 2(b) show the results of these causal mediation analyses when they were performed on both the aggregated attention head outputs and the MLP outputs (leaving the residual stream intact) at each sequence position and layer within the model. Consistent with our hypothesized architecture, the abstract causal mediation analysis (Figure 2(a)) revealed two distinct stages of processing, one in early layers of the model, with an effect that was largely concentrated at the positions of the final item in each in-context example, and one in intermediate layers, with an effect that was concentrated at the final position in the sequence (i.e., the position at which the model must generate a completion to the query). These results are consistent with the hypothesized behavior (both in terms of specific positions within the sequence, and relative order across layers) of symbol abstraction heads and symbolic induction heads respectively. The token causal mediation analysis (Figure 2(b)) revealed a later stage of processing that was also concentrated at the final position in the sequence, consistent with the hypothesized behavior of retrieval heads.

Next, we performed causal mediation analysis on the output of individual attention heads. To isolate symbol abstraction heads, we performed the abstract causal mediation analysis at the positions corresponding to the final item in each in-context example. To isolate symbolic induction heads, we performed the abstract causal mediation analysis at the final position in the sequence. To isolate retrieval heads, we performed the token causal mediation analysis at the final position in the sequence. The results (Figures 2(c)-2(e)) revealed a relatively sparse selection of attention heads, again conforming to the hypothesized three-stage structure.

### 3.2. Attention Analyses

We analyzed attention patterns to better understand the behavior of the identified attention heads. Figure 3(a) shows the attention patterns for symbol abstraction heads. Our hypothesis predicts that attention should be directed from the third item in each in-context example to the first item for ABA rules (top), and should be directed from the third item to the second item for ABB rules (bottom). The results largely confirmed this hypothesis (the attention patterns for these positions are highlighted with red dashed lines). Interestingly, the pattern became more focused for the second in-context example, suggesting that the symbol abstraction heads benefit from in-context learning.

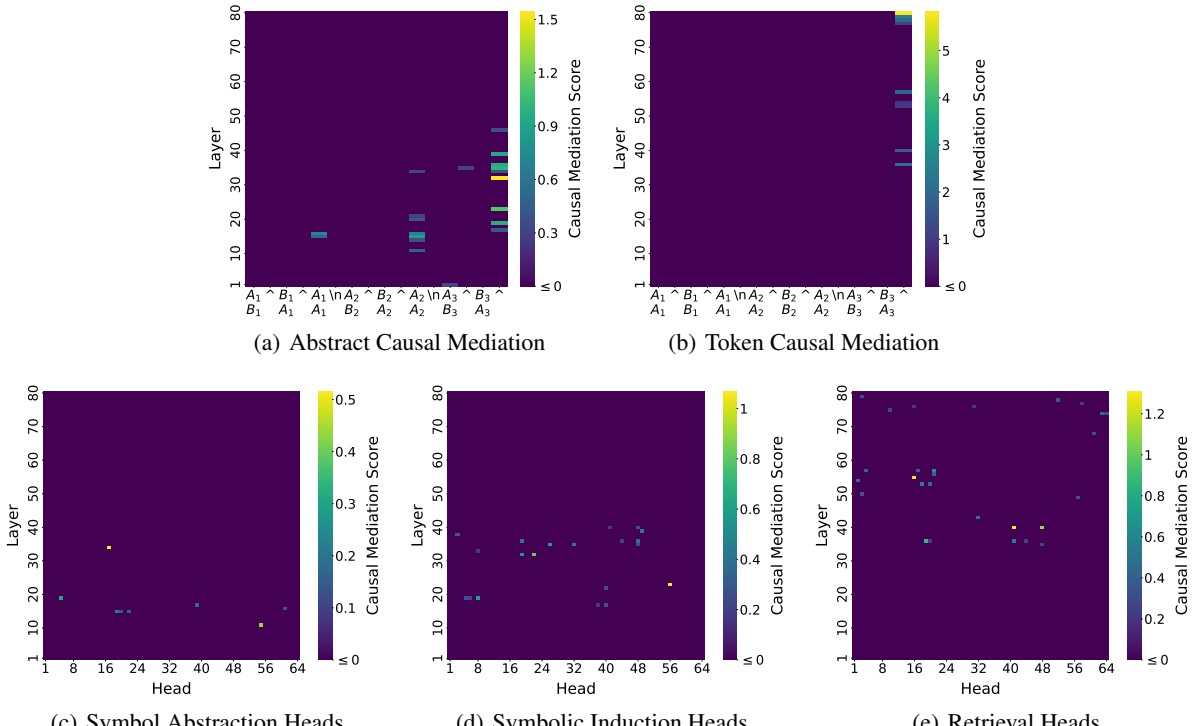

(a) Abstract Causal Mediation

(b) Token Causal Mediation

(c) Symbol Abstraction Heads

(d) Symbolic Induction Heads

(e) Retrieval Heads

*Figure 2.* **Causal Mediation Results. (a)** Abstract causal mediation effects at each layer and sequence position. X-axis shows the two contexts ($c_1^{abstract}$ and $c_2^{abstract}$) used for this analysis, aligned with each sequence position. **(b)** Token causal mediation effects at each layer and sequence position, with corresponding $c_1^{token}$ and $c_2^{token}$ contexts shown along X-axis. **(c)** Identification of symbol abstraction heads: abstract causal mediation effects for each attention head, averaged across positions corresponding to the last item in each of the two in-context examples. **(d)** Identification of symbolic induction heads: abstract causal mediation effects for each attention head, for the final token in the sequence. **(e)** Identification of retrieval heads: token causal mediation effects for each attention head, for the final token in the sequence. Permutation testing was performed to estimate the family-wise error rate across all scores in each plot, and scores were thresholded so that only scores significantly greater than zero ($p < 0.05$) are shown.

Figure 3(b) shows the results for symbolic induction heads. Our hypothesis predicts that attention should be directed from the final position in the entire sequence (the separation token at the end of the incomplete example) to positions of the final items in each in-context example, as these are the positions where the previous instances of the to-be-predicted abstract variable are located. The results confirmed this hypothesis. Again, the pattern was stronger for the second in-context example, consistent with a general effect of in-context learning.

Figure 3(c)) shows the results for retrieval heads. Our hypothesis predicts that attention should be directed from the final position in the sequence to the positions corresponding to the tokens that will appear next (i.e., within the incomplete example). For ABA rules (top), we predict that attention should be directed to the first item in the example (corresponding to the variable A), and for ABB rules (bottom) we predict that attention should be directed to the second item in each example (corresponding to the variable B). This prediction too was confirmed by our analyses.

Taken together, these results suggest that the model employs a form of indirection, first computing the variable associated with the predicted token, and then using that variable as a pointer to retrieve the token itself.

### 3.3. Representational Similarity Analyses

We also performed representational similarity analyses (Kriegeskorte et al., 2008) to better understand the representations that were produced by each type of attention head. In this analysis, representations are modeled in terms of their similarity with one another. For each set of tokens $A_1, B_1, ... A_N, B_N$, we created four prompts, intended to dissociate representations of abstract variables (i.e., symbols) vs. literal tokens. The first two prompts were the same as the $c_1^{abstract}$ and $c_2^{abstract}$ contexts used for causal mediation analysis, in which the same token plays different abstract roles. The other two prompts were based on $c_1^{abstract}$ and $c_2^{abstract}$, but the final instances of $A_N$ and $B_N$ were swapped. The resulting set of prompts predict one pattern of pairwise similarity for abstract variables, and a

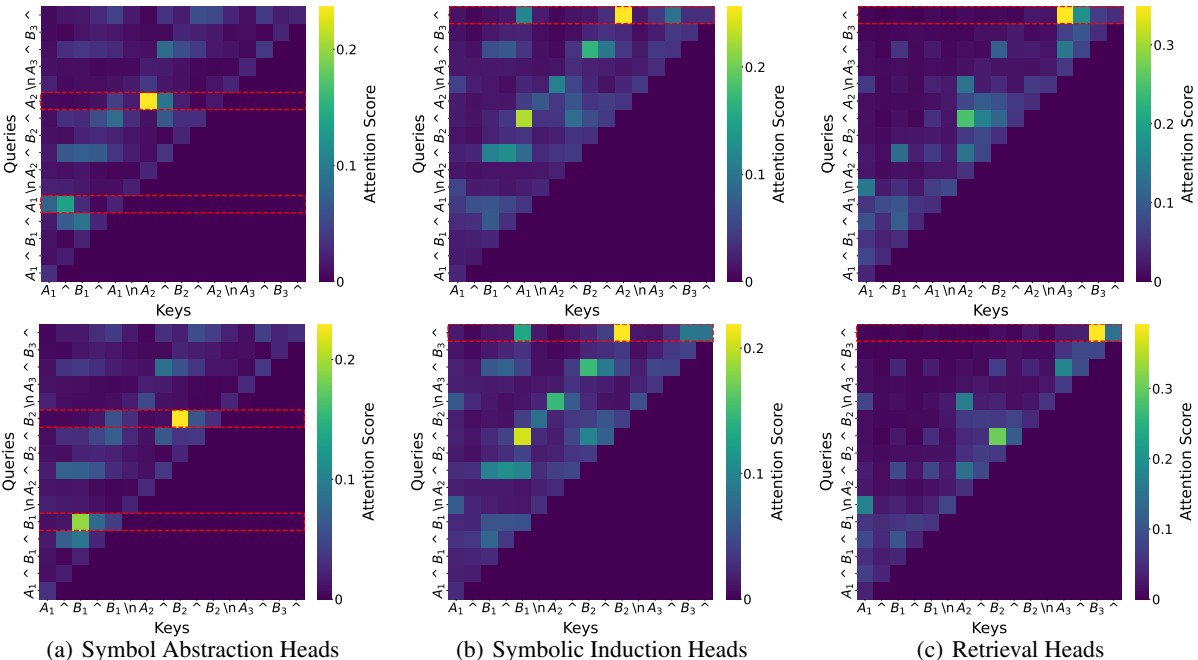

(a) Symbol Abstraction Heads      (b) Symbolic Induction Heads      (c) Retrieval Heads

*Figure 3.* **Attention Analysis.** Analysis of attention patterns for **(a)** symbol abstraction heads, **(b)** symbolic induction heads, and **(c)** retrieval heads. For each type of attention head, we selected only heads with statistically significant CMA scores (shown in Figure 2(c)-2(e)), and a weighted average of their attention patterns was computed using the causal mediation scores as weights. X-axis corresponds to keys (positions that are attended to), Y-axis corresponds to queries (positions from which attention is directed). Top row depicts attention pattern for ABA problems, bottom row depicts attention pattern for ABB problems. Prompt templates are shown along each axis, with tokens aligned to their corresponding positions. Red dashed lines highlight positions of interest (discussed in text). Note that beginning-of-sequence token is omitted.

different pattern of similarity for literal tokens.

Figure 4(a) shows the predicted pattern of pairwise similarities for representations of abstract variables, with all pairs involving two instances of the variable $A$ forming one block of high similarity, and all pairs involving two instances of $B$ forming another block. By contrast, Figure 4(b) shows the predicted pattern of pairwise similarities for representations of literal tokens. This pattern displays 3 diagonal bands, corresponding to pairs of the same token (regardless of whether this token is assigned to the same variable).

Figure 4(c) shows the pattern of similarity observed for the output of symbol abstraction heads at the third position in both in-context examples (averaged across the two examples). The pattern closely resembles the abstract variable similarity matrix, indicating that the output of symbol abstraction heads have a representational structure similar to abstract variables. Figure 4(d) shows the pattern observed for symbolic induction heads at the final sequence position. This pattern also resembles the pattern predicted for abstract variables. Figure 4(e) shows the patten observed for retrieval heads at the final sequence position. This pattern closely resembles the token similarity matrix.

Interestingly, although the output embeddings for symbol abstraction heads and symbolic induction heads primarily resemble the pattern predicted for abstract variables, they also show the diagonal bands predicted by the token similarity matrix, suggesting that they preserve some degree of token identity, and thus do not *exclusively* represent abstract variables. However, it is still possible for abstract variables to be represented in an invariant manner in a *subspace* of these attention heads' outputs. To test this, we performed a decoding analysis in which a decoder was trained to predict the abstract rule (ABA vs. ABB) for problems involving one set of tokens, and then tested on problems involving a completely different set of tokens. This decoding analysis yielded nearly perfect generalization accuracy for both symbol abstraction and symbolic induction heads ($> 98\%$ test accuracy, see Section C.7 for more details), indeed suggesting that abstract variables are invariantly represented in a subspace of the output embeddings for these heads.

To gain a more precise understanding of the identified attention heads, we also applied RSA to the key, query, and value embeddings (Tables 2-3 and Figures 11-14). For symbol abstraction heads, we found that queries primarily represented token identity, keys represented a mixture of both tokens

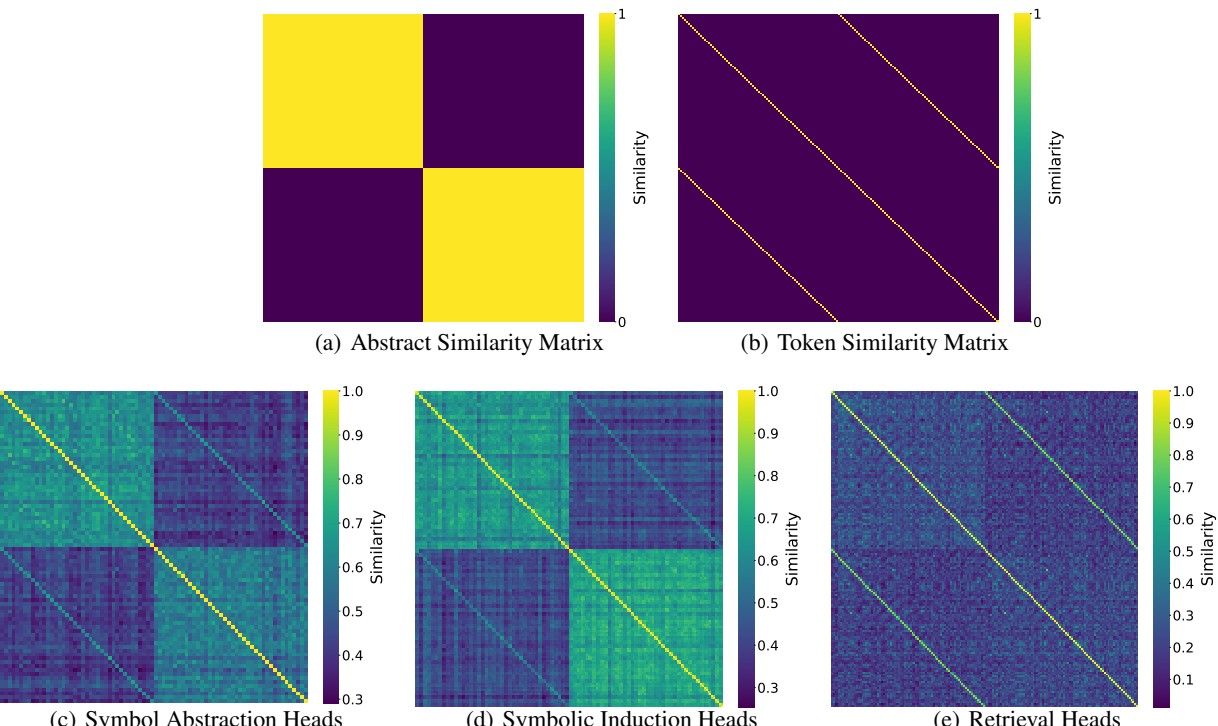

*Figure 4.* **Representational Similarity Analysis. (a)** Predicted pattern of pairwise similarity for representations of abstract variables. **(b)** Predicted pattern for representations of tokens. **(c)** Pairwise similarity for symbol abstraction head outputs, averaged across the third position in the two in-context examples. **(d)** Pairwise similarity for symbolic induction head outputs at the final sequence position. **(e)** Pairwise similarity for retrieval head outputs at the final sequence position. For each attention head type, we computed a weighted average of similarity matrices from all statistically significant heads, using the causal mediation scores as weights.

and abstract variables, and values primarily represented the abstract variables. For symbolic induction heads, queries and keys primarily represented the relative position within each in-context example, while values primarily represented abstract variables. For retrieval heads, queries primarily represented abstract variables, keys represented a mixture of both tokens and variables, and values primarily represented the predicted token. These results further confirm the hypothesized mechanisms, namely that abstraction heads convert tokens to variables, symbolic induction heads make predictions over these variables, and retrieval heads convert variables back to tokens.

### 3.4. Ablation Analyses

The causal mediation analyses in section 3.1 demonstrate that the identified attention heads are causally sufficient, in the sense that perturbing their outputs alters the model's responses in a predictable manner. We also performed ablation analyses to test whether these heads are *necessary* for the model to perform the task.

For each type of attention head, we performed a cumulative ablation analysis in which the heads with the top $h$ causal

mediation scores were ablated. This was performed for the full range of $h = 1...H$ heads in the entire model. We performed a control experiment in which each head ablated in the previous experiment was replaced by the head in the same layer with the *lowest* causal mediation score. Additionally, we implemented a random baseline by ablating $h$ randomly selected attention heads. We measured the effects of these ablations in terms of the probability assigned to the correct answer.

We found that these ablation experiments had a dramatic effect for all three types of heads (Figures 15(a)-15(c)). In the ablation condition, the probability assigned to the correct answer rapidly fell to zero as more heads were ablated, whereas in the control conditions it was necessary to ablate almost all attention heads to have such an effect. These experiments confirmed that all three types of attention heads were both sufficient and necessary to perform the rule induction task.

### 3.5. Comparison with Induction Heads

We investigated the relationship between symbolic induction heads and the standard induction heads identified in previous

work (Olsson et al., 2022). In that work, it was proposed that induction heads not only perform literal sequence induction, but may also perform a fuzzy or abstract form of induction. This raises the question of whether symbolic induction heads are merely standard induction heads.

For each attention head, we computed the prefix matching score previously used to identify induction heads (Olsson et al., 2022), and compared this with the causal mediation score for symbolic induction heads. We found that these scores were very weakly correlated ($r = 0.11$, Figure 9(a)). These results suggest that, despite the conceptual similarity between these two mechanisms, they are implemented by disjoint sets of attention heads.

### 3.6. Comparison with Function Vectors

We also investigated the relationship between symbolic induction heads and function vectors (Todd et al., 2024), representations of an in-context task that are generated by a subset of attention heads. The symbolic induction heads identified in the present work have many similarities to the attention heads that generate function vectors, including: 1) they are found in intermediate layers of the model, 2) they primarily attend to the last item in each in-context example, and 3) they are causally implicated in in-context learning for relational tasks.

To address this, we computed the average indirect effect for each attention head, which represents a measure of the extent to which its outputs constitute function vectors (Todd et al., 2024), and compared this with the causal mediation score for symbolic induction heads. This analysis revealed that these scores are indeed highly correlated (Figure 9(b); $r = 0.86$), suggesting that they are essentially the same set of attention heads. That is, the output of symbolic induction heads can be thought of as function vectors. This result provides a novel perspective on function vectors, suggesting that, where relevant, they can be used to implement symbolic forms of computation. This result also provides insight into the mechanism that computes function vectors, suggesting that it may be conceptualized as a form of sequence induction over abstract variables. Interestingly, this is consistent with recent findings that function vector heads often develop out of induction heads (Yin & Steinhardt, 2025).

Additionally, we found that these heads are not the only mechanism involved in computing function vectors. Specifically, we found that symbol abstraction heads also play an important role. The previous analysis was based on function vector scores computed at the the final position in the sequence. When function vector scores were instead computed based on the third item in each context-example (Figure 10), these scores were highly correlated with symbol abstraction heads ($r = 0.47$), but *not* symbolic induction heads ($r = 0.06$). These results suggest that function vec-

tors are first computed by symbol abstraction heads at the level of individual in-context examples, and symbolic induction heads are primarily responsible for aggregating them across in-context examples.

### 3.7. Evaluating Other Language Models

To determine how widespread the identified mechanisms are, we applied our causal mediation analyses to 12 additional open-source language models of different sizes across four model families: GPT-2, Gemma-2, Qwen2.5, and Llama 3.1 (Figure 16). With the exception of the GPT-2 models, all tested LLMs displayed nearly perfect accuracy on the identity rules task with 10 in-context examples. Correspondingly, all of the LLMs that performed well on the identity rules task also displayed statistically significant scores for each type of attention head, and these heads were arranged in a three-stage hierarchy (Figure 16(b) and Figures 20-22). By contrast, the GPT-2 variants did not show robust evidence for the presence of symbol abstraction heads (Figure 16(b) and 18), consistent with their poor performance on this task. These results further confirm the link between the identified mechanisms and abstract reasoning in LLMs, and suggest that symbolic mechanisms may only emerge at certain scales (in terms of model or training data size).

### 3.8. More Complex Reasoning Tasks

Our results thus far have focused on a simple but paradigmatic case of abstract reasoning – algebraic rule induction. Do these results also extend to more complex reasoning tasks? To test this, we extended our causal mediation analyses to two other abstract reasoning tasks (Figure 23). First, we investigated letter string analogies involving either successor or predecessor relations (Hofstadter et al., 1995). Second, we investigated verbal analogies involving either antonym or synonym relations. We found that the same basic three-stage architecture was present for both of these tasks (Figures 24 and 25), suggesting that similar mechanisms are indeed involved in more complex abstract reasoning tasks.

Interestingly, the causal mediation scores revealed a complex pattern of within- and between-task correlations. For the identity rules task and letter string analogies, we found moderate to high correlations both at the within-task level (i.e., for different relations or rules in the same task; Table 4) and the between-task level (Table 5). However, for verbal analogies, the correlations (both within- and between-task) were lower or even negative. These results suggest that, although all three tasks employed the same high-level mechanistic strategy, there was some specialization in the specific tasks and relations mediated by each attention head.

## 4. Related Work

There is a rich history of work illustrating how various aspects of symbol processing might be implemented in neural networks. Work on the tensor product representation (Smolensky, 1990) and binding-by-synchrony (Hummel & Holyoak, 2003) illustrated how dynamic variable-binding can be performed in neural networks. Kriete et al. (2013) demonstrated how indirection, the use of one variable to refer to another, can be implemented in a biologically plausible neural network. More recently, a series of studies illustrated how a *relational bottleneck* (Webb et al., 2024)–a strong inductive bias to perform relational processing–can enable data-efficient learning of abstract reasoning capabilities in deep learning systems (Webb et al., 2021; Kerg et al., 2022; Altabaa et al., 2024). The primary contribution of our work, relative to these previous studies, is to demonstrate empirically that symbolic mechanisms can emerge in a large-scale neural network, and to illustrate how they operate to support abstract reasoning. Notably, the symbol abstraction heads identified in this work implement an emergent version of the abstractor architecture that was previously proposed to support relational learning (Altabaa et al., 2024)

There has also been much recent work investigating the internal mechanisms that support various forms of abstract and structured task processing in large language models. This work has identified key primitives such as induction heads (Olsson et al., 2022), function vectors (Todd et al., 2024), concept vectors (Opiełka et al., 2025), binding IDs (Feng & Steinhardt, 2024), and other mechanisms that play a role in relational processing (Merullo et al., 2023). Complementary work has also investigated how structured mechanisms emerge in smaller models trained on synthetic tasks (Wu et al., 2025; Grant et al., 2025; Al-Saeedi & Härmä, 2025; Tang et al., 2025; Brinkmann et al., 2024). We build on this previous work by identifying an integrated architecture that brings together multiple mechanisms. These include newly identified mechanisms – symbol abstraction and symbolic induction heads – that, respectively, carry out the processes of abstraction and rule induction needed to implement an emergent form of symbol processing that supports abstract reasoning in neural networks.

## 5. Discussion

In this work, we have identified an emergent architecture consisting of several newly identified mechanistic primitives, and illustrated how these mechanisms work together to implement a form of symbol processing. These results have major implications both for the debate over whether language models are capable of genuine reasoning, and for the broader debate between traditional symbolic and neural network approaches in artificial intelligence and cognitive science.

On the one hand, the emergent architecture identified here, that supports abstract reasoning via an intermediate layer of symbol processing, is strikingly at odds with characterizations of language models as mere stochastic parrots (Bender et al., 2021) or 'approximate retrieval' engines (Wu et al., 2024). These results are also at odds with claims that neural networks will need innately configured symbol processing mechanisms in order to perform human-like abstract reasoning (Marcus, 2001; Dehaene et al., 2022; Wong et al., 2023). On the other hand, these results can be viewed as a vindication of longstanding claims that symbol-processing mechanisms of some form (whether they be innate or learned) are a necessary component supporting human cognitive abilities (Fodor & Pylyshyn, 1988), insofar as they suggest that neural networks can acquire these abilities only by developing their own form of symbol processing.

It is interesting to consider the extent to which the identified mechanisms are truly emergent vs. dependent on innate aspects of the model. The transformer architecture (Vaswani et al., 2017) does not obviously possess the strong relational inductive biases that characterize the abstractor (Altabaa et al., 2024) or other architectures designed to perform relational abstraction (Webb et al., 2024). However, transformers do have some inductive biases that seem relevant, including: 1) an innate mechanism for computing in-context similarity via the inner product between keys and queries, and 2) a form of indirection, in the sense that the keys and queries that are used to select information for retrieval are distinct from the values that are retrieved. In future work, it would be interesting to investigate the extent to which these or other inductive biases contribute to the development of emergent symbol processing mechanisms.

Finally, an important open question concerns the extent to which language models precisely implement symbolic processes, as opposed to merely approximating these processes. In our representational analyses, we found that the identified mechanisms do not *exclusively* represent abstract variables, but rather contain some information about the specific tokens that are used in each problem. On the other hand, using decoding analyses, we found that these outputs contain a subspace in which variables are represented more abstractly. A related question concerns the extent to which *human* reasoners employ perfectly abstract vs. approximate symbolic representations. Psychological studies have extensively documented 'content effects', in which reasoning performance is not entirely abstract, but depends on the specific content over which reasoning is performed (Wason, 1968), and recent work has shown that language models display similar effects (Lampinen et al., 2024). In future work, it would be interesting to explore whether such effects are due to the use of approximate symbolic mechanisms, and whether similar mechanisms are employed by the human brain.

## Impact Statement

This paper presents work whose goal is to advance the field of Machine Learning. There are many potential societal consequences of our work, none which we feel must be specifically highlighted here.

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

# A. Code and Hardware

All code was written in Python using the TransformerLens and HuggingFace libraries. Experiments on Llama-3.1 70B and Qwen2.5 72B were conducted on two NVIDIA 80G H100 GPUs while experiments on other models of smaller sizes were conducted on a single H100 GPU. All model weights are loaded in the bfloat16 format.

All code and data will be released at https://github.com/yukang123/LLMSymbMech.

# B. Implementation Details

## B.1. Rule Induction Task: Evaluating Task Performance

To evaluate performance on the rule induction task, we randomly selected English tokens from the LLama-3 vocabulary to form 2,000 prompts. We used the following 2-shot prompt format:

$$A_1\hat{\ }B_1\hat{\ }A_1\backslash nA_2\hat{\ }B_2\hat{\ }A_2\backslash n\ A_3\hat{\ }B_3\hat{\ }$$

## B.2. Causal Mediation Analysis

---

**Algorithm 1** Causal Mediation Analysis

---

**Input:** context pair $(c_1, c_2)$; language model $f(c) \in R^A$: outputs the logits for all possible next tokens at the last position of prompt $c$; vocabulary size $A$; the $i$-th value of vector $f(c)$: $f(c)[i]$; the correct answer for $c_1$: $y_{c_1}$; the expected answer for the patched context $c_1^*$: $y_{c_1^*}$.

1: For $c_1$, measure the difference between the output logit for $y_{c_1^*}$ and $y_{c_1}$, i.e., $\Delta f_{c_1} = f(c_1)[y_{c_1^*}] - f(c_1)[y_{c_1}]$.
2: Cache the internal activations after feeding $c_2$ into the model.
3: Replace activations for selected model component (e.g., output of the entire attention block, or individual attention head, at a particular layer and token position) in $c_1$ with corresponding activations in $c_2$, yielding patched context $f_{c_1^*}$.
4: Compute logit difference for patched context:

$$\Delta f_{c_1^*} = f(c_1^*)[y_{c_1^*}] - f(c_1^*)[y_{c_1}] \tag{6}$$

5: Compute causal mediation score:

$$s = \Delta f_{c_1^*} - \Delta f_{c_1} = (f(c_1^*)[y_{c_1^*}] - f(c_1^*)[y_{c_1}]) - (f(c_1)[y_{c_1^*}] - f(c_1)[y_{c_1}]) \tag{7}$$

**Output**: Causal Mediation Score $s$

---

Algorithm 1 explains the causal mediation procedure for a given context pair $(c_1, c_2)$. We performed three separate versions of this analysis, each targeting one of the hypothesized attention head mechanisms, as described below.

**Targeting Symbol Abstraction Heads** To target the hypothesized symbol abstraction heads, we used the context pair $(c_1^{abstract}, c_2^{abstract})$. In one version of this analysis, $c_1^{abstract}$ involved an ABA rule, and $c_2^{abstract}$ involved an ABB rule. Importantly, the same tokens were used for both contexts, but with swapped variable assignments, such that the correct answer to complete both contexts was the same. We also performed a version of this analysis in which $c_1^{abstract}$ was an ABB rule and $c_2^{abstract}$ was an ABA rule. We used 100 prompts for each version of the analysis, yielding 200 prompts in total. Patching was performed at the position of the final token in each of the two in-context examples. See Figure 5 for an illustration and further explanation of this analysis.

**Targeting Symbolic Induction Heads.** To target the hypothesized symbolic induction heads, we performed the same analysis described above (for symbol abstraction heads), except that patching was performed at the final position in the sequence, i.e. the position at which the model was expected to generate the missing token. See Figure 6 for an illustration of this analysis.

**Targeting Retrieval Heads.** To target retrieval heads, we used the context pair $(c_1^{token}, c_2^{token})$. In one version of this analysis, $c_1^{token}$ involved an ABA rule, and $c_2^{token}$ involved the same ABA prompt, but with the position of the final two tokens swapped. We also performed a version of this analysis in which both $c_1^{token}$ and $c_2^{token}$ involved an ABB rule. Patching was performed at the final position in the sequence. See Figure 7 for an illustration of this analysis.

**Activations.** We performed CMA on two types of activations: 1) the output of the entire attention block (including both the attention head and MLP outputs) at a specific layer and token position, and 2) the output of a specific attention head (before the additional projection matrix over concatenated head outputs) at a specific layer and token position. The first analysis was intended to provide a course-grained, but comprehensive, account of where the causally relevant computations were located (i.e., which layers and token positions). The second analysis was intended to provide a more fine-grained analysis (focusing on individual attention heads) of the causally relevant model components at those locations.

**Permutation Testing.** We performed statistical tests to identify significant heads for each head type. Specifically, we used permutation testing to model the null distribution for these analyses. For each attention head $h$, the average causal score over multiple context pairs measures the difference between the means of two groups, i.e., the logit difference for the original context $\Delta f_{c_1}$ and the logit difference for the patched context $\Delta f_{c_1^*}$. To model the null distribution, we randomly permuted these two quantities, such that there was a 50% chance they would remain the same, and a 50% chance $\Delta f_{c_1}$ would be swapped with $\Delta f_{c_1^*}$ and vice versa. This was repeated 5,000 times. We then identified the threshold $\epsilon$ for which there was a family-wise error rate of $p < 0.05$. That is, we identified the threshold for which there was a less than 5% chance that even a single attention head would have a causal mediation score exceeding this threshold in the null distribution. We considered only heads with scores that passed this threshold to be statistically significant.

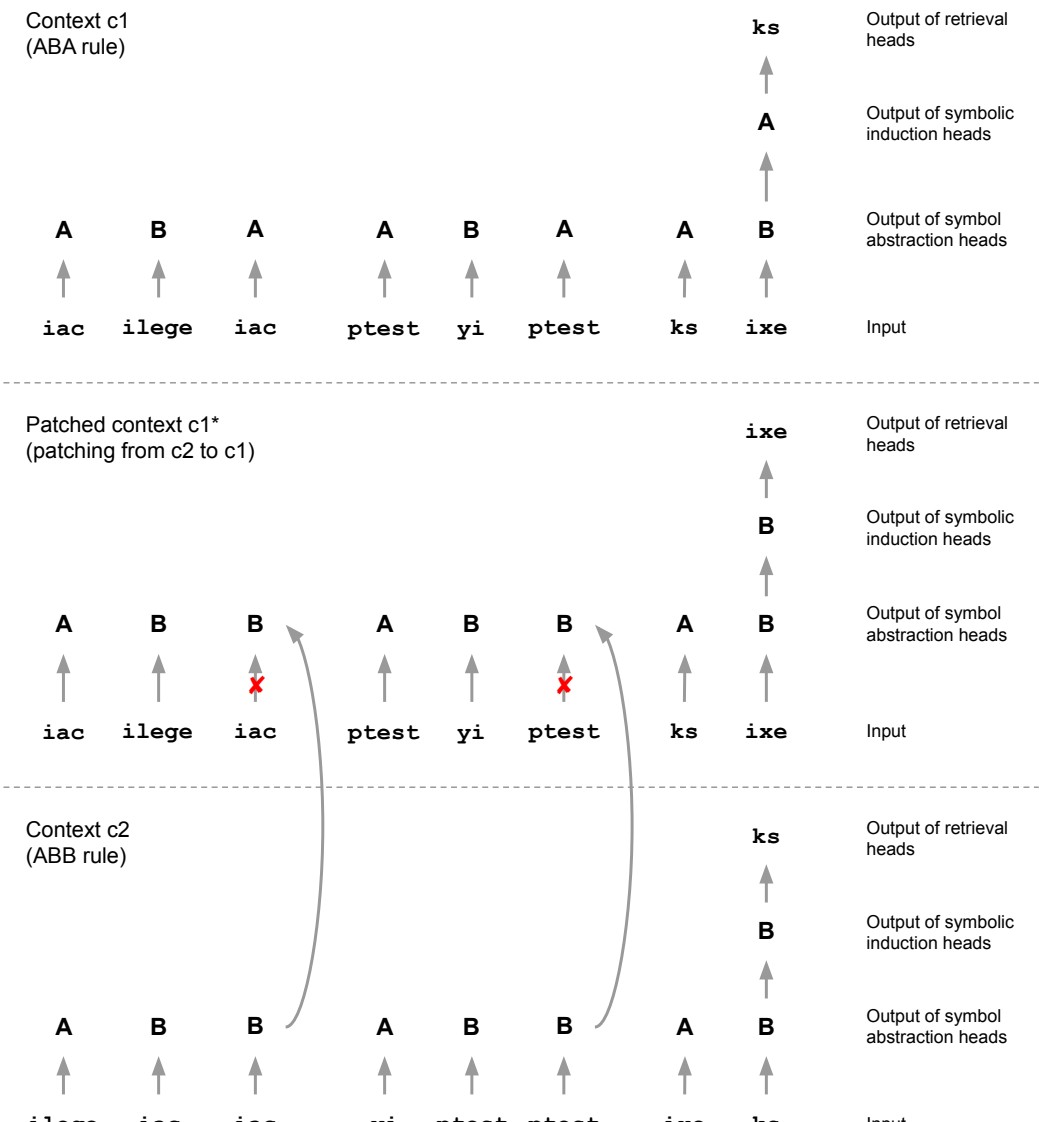

*Figure 5.* **Illustration of Causal Mediation Analysis targeting Symbol Abstraction Heads**. In this example, context $c_1$ involves an ABA rule, and context $c_2$ involves an ABB rule. Importantly, both contexts are formed from the same token sets, but with the tokens assigned to different variables, such that the correct prediction for the next token is the same in both contexts ('ks'). Patching from context $c_2$ to $c_1$ should therefore not have an effect on model components that encode literal tokens (i.e., the retrieval heads), since the output of these components should be the same in both contexts. This intervention should, however, have an effect on model components that encode abstract variables, since these differ between the two contexts. To target symbol abstraction heads, we patched from $c_2$ to $c_1$ at the position of the final tokens in each in-context example, thus intervening on the abstract variable representation for these tokens (and therefore converting the representation of an ABA rule into a representation of an ABB rule). In this example, the correct answer $y_{c_1}$ for context $c_1$ is 'ks' (i.e., $A_N$), and the expected answer $y_{c_1^*}$ for the patched context $c_1^*$ is 'ixe' (i.e., $B_N$).

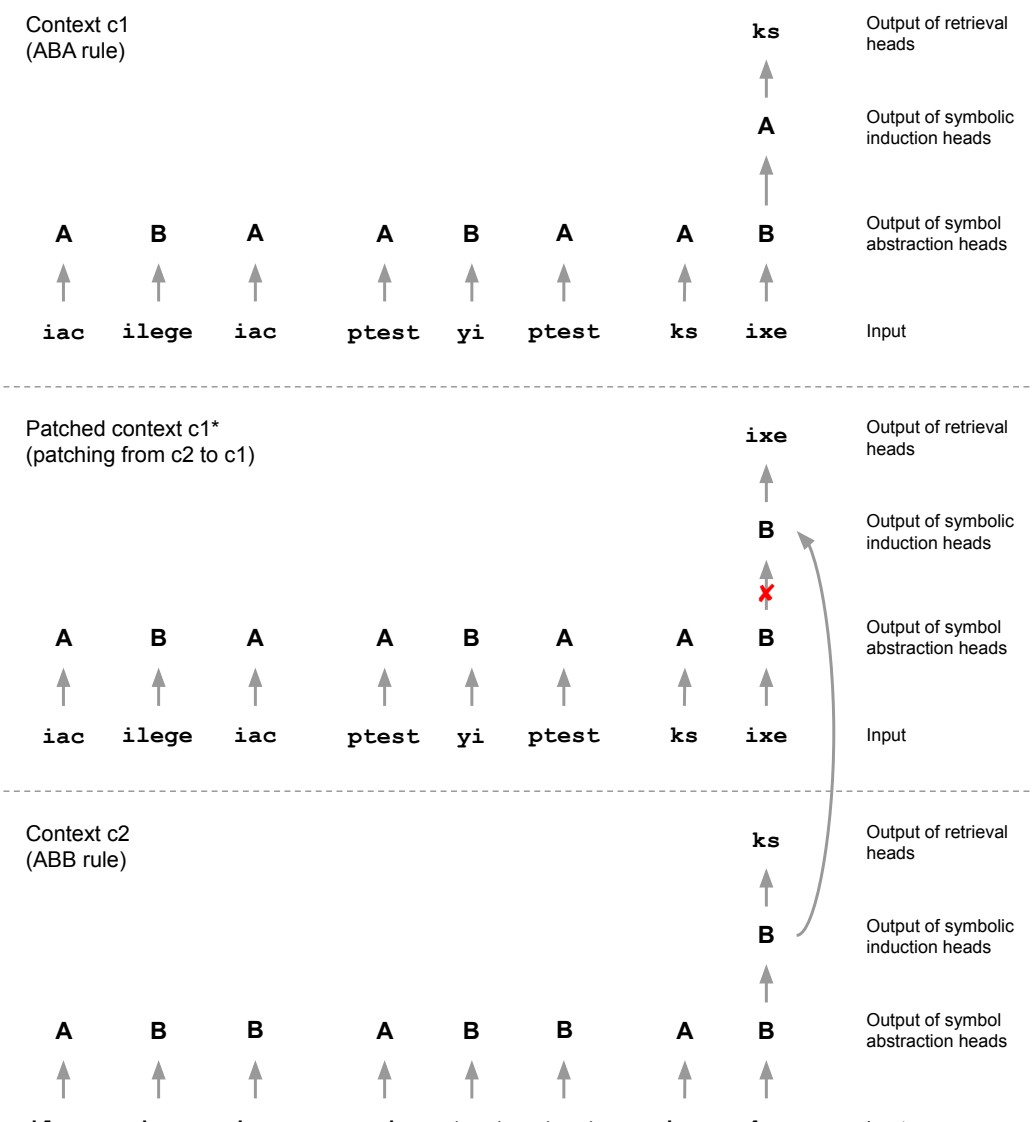

*Figure 6.* **Illustration of Causal Mediation Analysis targeting Symbolic Induction Heads**. The intervention targeting symbolic induction heads was the same as the intervention targeting symbol abstraction heads, except that patching was performed at the final position in the sequence, at which the model was expected to generate the final token. This intervention should have an effect on the process of predicting the abstract variable associated with the final token, but not on the process of retrieving the final token itself.

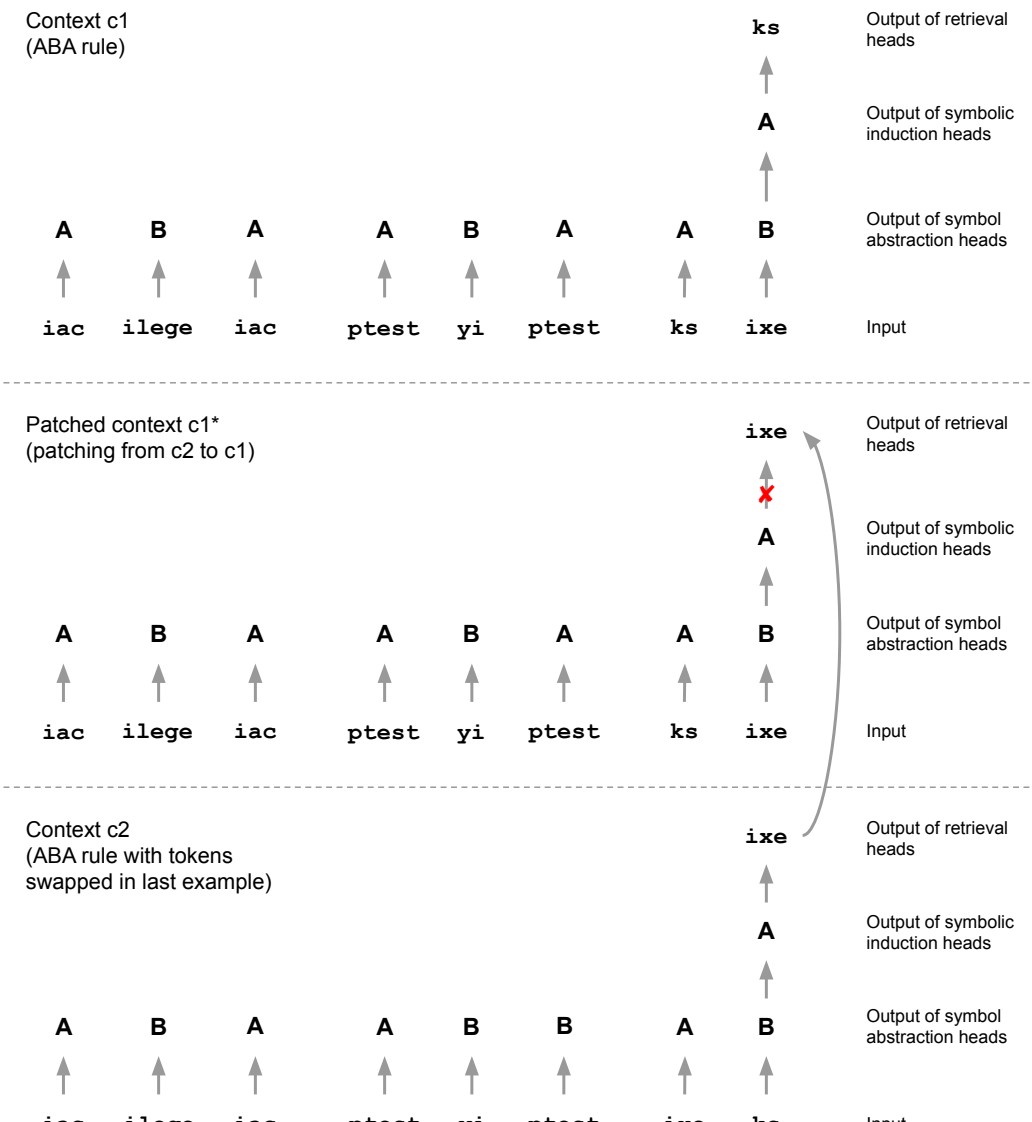

*Figure 7.* **Illustration of Causal Mediation Analysis targeting Retrieval Heads**. In this example, both contexts $c_1$ and $c_2$ involve an ABA rule, but the order of the tokens in the final in-context examples are swapped. This means that the correct answer differs between these two contexts, and therefore model components that encode literal tokens such as retrieval heads should be affected by this intervention. Model components that encode abstract variables, however, should not be affected by this intervention, since the abstract rule in both contexts is the same. To target Retrieval Heads, we patched from $c_2$ to $c_1$ at the final position in the sequence. In this example, the correct answer $y_{c_1}$ for context $c_1$ is 'ks' (i.e., $A_N$), and the expected answer $y_{c_1^*}$ for the patched context $c_1^*$ is 'ixe' (i.e., $B_N$).

## B.3. Attention Analyses

For each individual attention head, the attention map was averaged over 1,378 prompts each for the ABA and ABB tasks, again limiting the analysis to prompts that the model answered correctly.

## B.4. Representational Similarity Analyses

For one set of tokens $[(A_n, B_n)]_{n=1..N}$, we built four different contexts, i.e.,

$$A_1, B_1, A_1, ..., A_N, B_N \qquad (8)$$
$$A_1, B_1, A_1, ..., B_N, A_N$$
$$B_1, A_1, A_1, ..., B_N, A_N$$
$$B_1, A_1, A_1, ..., A_N, B_N$$

We randomly selected 40 token sets formed into the above contexts to measure the similarity of attention head activations at certain token positions. We then compared these similarity matrices with hypothesized similarity matrices based on either abstract variables or token identities. We applied this analysis to four separate attention head components: keys, values, queries, and outputs. Figure 4 shows the RSA on the output of significant heads for each head type.

For symbol abstraction heads, we measured query similarity and output similarity at the third item in the first two in-context examples, and keys and values at the first two items. For symbolic induction heads, we compared the values at the third item in the first two in-context examples and the outputs at the final position of the prompt. For retrieval heads, we used the queries and outputs at the final sequence position and keys and values at the first two items in the last incomplete query. These positions were chosen because they most directly targeted the hypothesized computations in the proposed three-stage architecture.

For the keys and queries of symbolic induction heads, we tested two hypotheses. Inspired by the original induction head hypothesis (Olsson et al., 2022), the first hypothesis is that these embeddings represent the abstract variable at the previous position (just as the key and query embeddings in induction heads represent the token at the previous position). However, this mechanism is limited to performing induction via bigram statistics. An alternative hypotehsis is that these embeddings represent the relative position within an in-context example. This mechanism can support induction based on more complex n-gram statistics. Our original task, involving either ABA or ABB rules, does not enable us to distinguish between these two hypotheses, as both hypotheses predict the same similarity matrix. To address this, we tested a variant of this task in which the two hypotheses dissociate. Specifically, we used a task involving the following length-4 identity rules: AABA, ABCB, and ABCC. We performed RSA based on the similarity of the embeddings at the third and fourth items in each in-context example.

## B.5. Ablation Analyses

We randomly selected a set of 40 prompts that do not overlap with the prompts used in causal mediation analyses. Starting from the heads with highest causal mediation scores, we gradually increased the number of ablated heads, and evaluated the effect on the probability of the correct answer. As a control, we performed the same analysis, but replaced each ablated head with the head in the same layer with the lowest causal mediation score. We also included a random baseline by randomly selecting heads to ablate. The results in Figure 15 are averaged over both ABA and ABB tasks.

## B.6. Induction Heads

Following (Olsson et al., 2022), we used the prefix matching score as a measure for induction heads. We created a prompt involving a repeated sequence of 50 random tokens. The prefix matching score is defined as the average attention score from each token to the tokens that directly follow the previous instance of the same token. We averaged results over 4 random seeds.

### B.7. Function Vectors

As described in (Todd et al., 2024), function vectors are aggregated over heads with a high causal mediation score. The answers for each in-context example are shuffled to form a *corrupted* prompt and the causal indirect effect (CIE) is defined as the recovery of the probability for correct answers after replacing the attention head output at the final position in the corrupted run with the average embedding across multiple prompts from the clean run. The average indirect effect (AIE) was taken over 50 prompts each for ABA and ABB tasks.

## C. Additional Experimental Results

### C.1. Comparison of Symbol Abstraction Heads, Symbolic Induction Heads, and Retrieval Heads

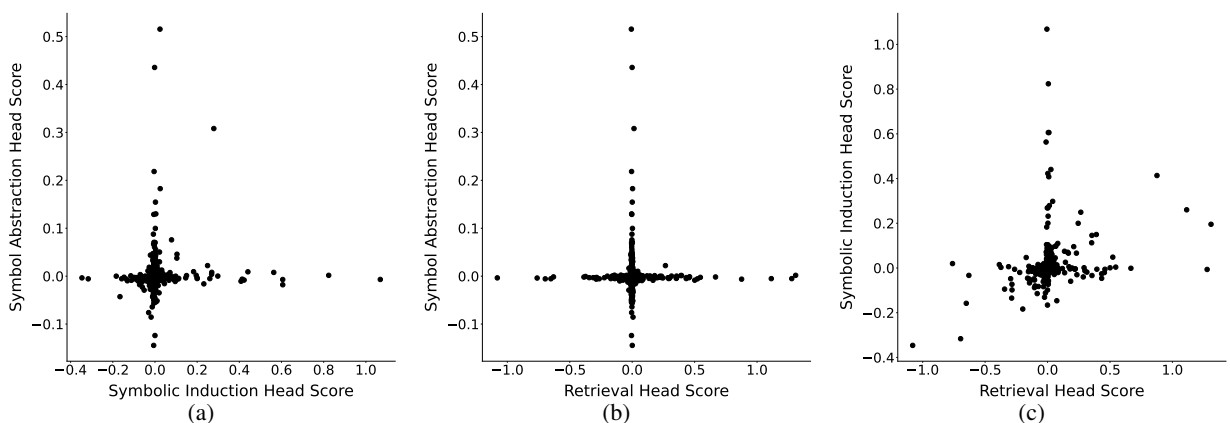

*Figure 8.* **(a)** Comparison of symbol abstraction head scores and symbolic induction head scores. **(b)** Comparison of symbol abstraction head scores and retrieval head scores. **(c)** Comparison of symbolic induction head scores and retrieval head scores. Each dot represents a single attention head, with all heads across all layers displayed. The scores for each type of attention head are largely orthogonal to each other, indicating that they form disjoint sets of attention heads.

## C.2. Comparison with Induction heads and Function Vectors

We compared the symbolic induction head scores with prefix matching scores, used as a measure for standard induction heads (Olsson et al., 2022), and average indirect effects, used as a measure for function vectors (Todd et al., 2024), across all attention heads (Figure 9).

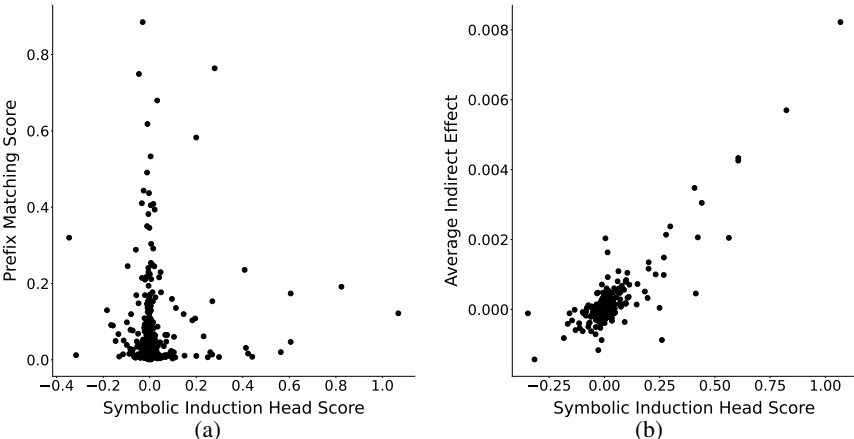

(a)          (b)

*Figure 9.* (a) Comparison of symbolic induction heads and standard induction heads. (b) Comparison of symbolic induction heads and function vector attention heads.

Apart from the original implementation in (Todd et al., 2024) on the embeddings at the final sequence position (Section B.7), we also computed the function vector scores by patching the activations at the position of the third item in each in-context example. Table 1 and Figure 10 show the result of comparing function vector scores based on either of these two positions (last position in sequence vs. position of third item in each in-context example) to the scores for symbol abstraction heads and symbolic induction heads. The results indicate that symbol abstraction head scores are correlated with function vector scores at the third item in each in-context example, while symbolic induction head scores are correlated with function vector scores at the final position in the sequence.

|  | Symbol Abstraction Head Score | Symbolic Induction Head Score |
|---|---|---|
| Third-item-position Function Vector Score | 0.47 | 0.06 |
| Final-position Function Vector Score | 0.06 | 0.86 |

*Table 1.* **Comparison between Function Vectors and Symbol Abstraction Heads/Symbolic Induction Heads.** Function vector scores (average indirect effect) are computed based on embeddings either at the third item in each in-context example, or at the final position in the sequence (original implementation in (Todd et al., 2024)). Results reflect the correlation coefficient for comparisons between function vector scores and symbol abstraction head and symbolic induction head scores. Corresponding scatter plots are shown in Fig. 10.

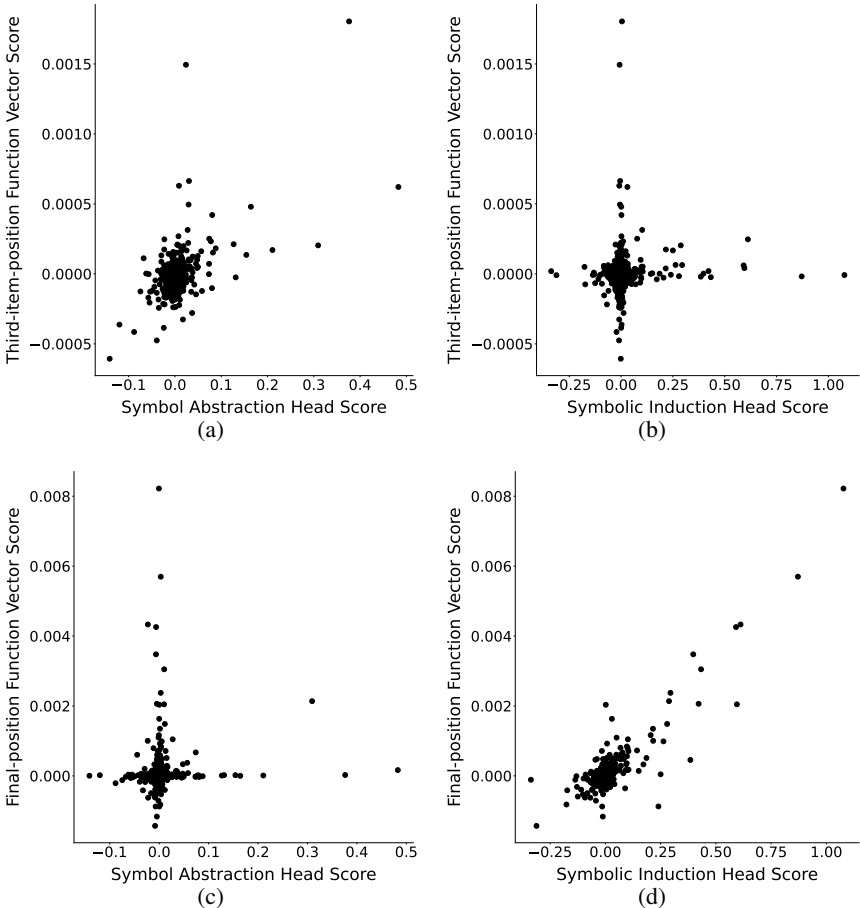

*Figure 10.* **Scatter Plots for Comparisons between Function Vectors and Symbol Abstraction Heads/Symbolic Induction Heads.** Each dot represents a single attention head, with all heads across all layers displayed. (a) Symbol abstraction head scores were correlated with function vector scores evaluated at the third items in the in-context examples, but (c) not at the final position. (d) Symbolic induction head scores were highly correlated with function vector scores evaluated at the final position, but (b) not at the third items in the in-context examples.

## C.3. Representational Similarity Analyses for Different Attention Head Components

Figures 4-14 show the hypothesized similarity matrices and the actual similarity matrices computed from the key, query, value, and output embeddings. For illustration purposes, we used only 10 different token sets to compute the similarity in Figures 11-14.

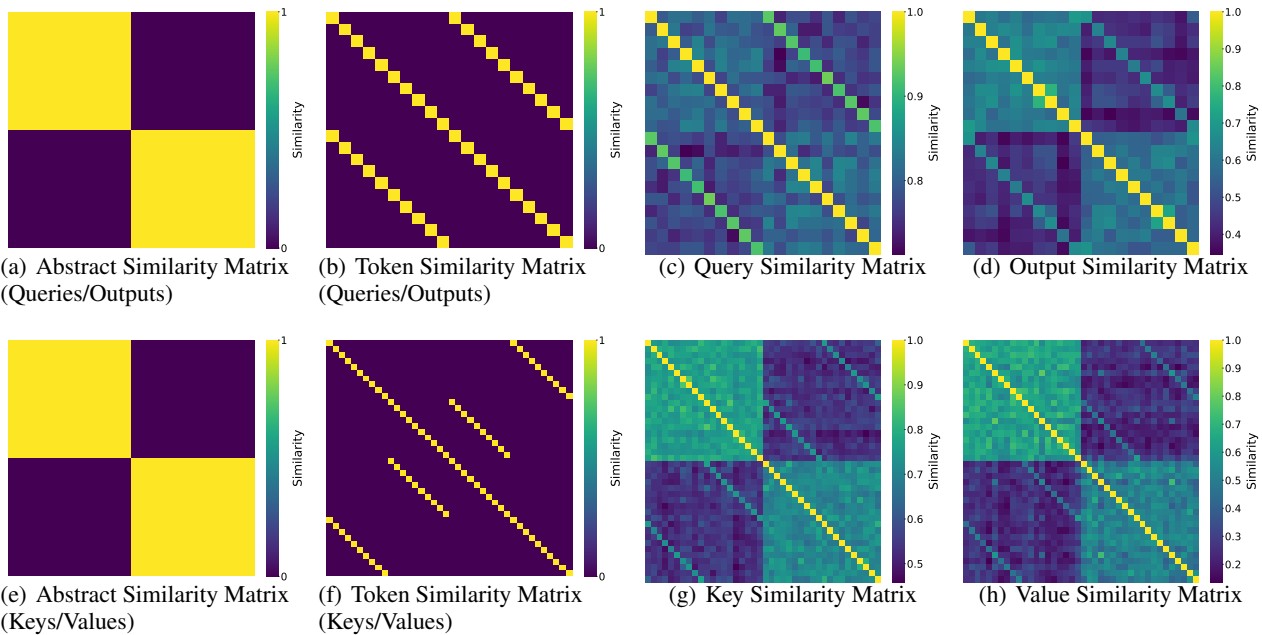

(a) Abstract Similarity Matrix (Queries/Outputs)

(b) Token Similarity Matrix (Queries/Outputs)

(c) Query Similarity Matrix

(d) Output Similarity Matrix

(e) Abstract Similarity Matrix (Keys/Values)

(f) Token Similarity Matrix (Keys/Values)

(g) Key Similarity Matrix

(h) Value Similarity Matrix

*Figure 11.* **Representational Similarity Analysis for Symbol Abstraction Heads.** (a), (b), (e), and (f) are the predicted abstract or token similarity matrices; (c), (d), (g), and (h) are the actual embedding similarity matrices.

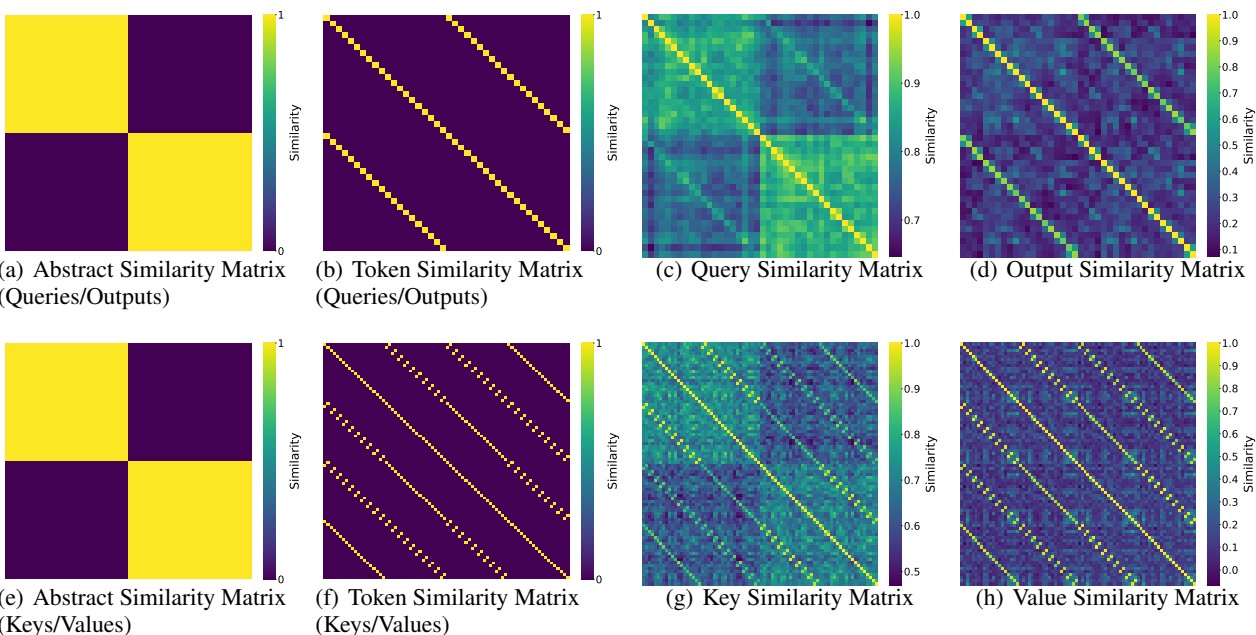

(a) Abstract Similarity Matrix (Queries/Outputs)

(b) Token Similarity Matrix (Queries/Outputs)

(c) Query Similarity Matrix

(d) Output Similarity Matrix

(e) Abstract Similarity Matrix (Keys/Values)

(f) Token Similarity Matrix (Keys/Values)

(g) Key Similarity Matrix

(h) Value Similarity Matrix

*Figure 12.* **Representational Similarity Analysis for Retrieval Heads.** (a), (b), (e), and (f) are the predicted abstract or token similarity matrices; (c), (d), (g), and (h) are the actual embedding similarity matrices.

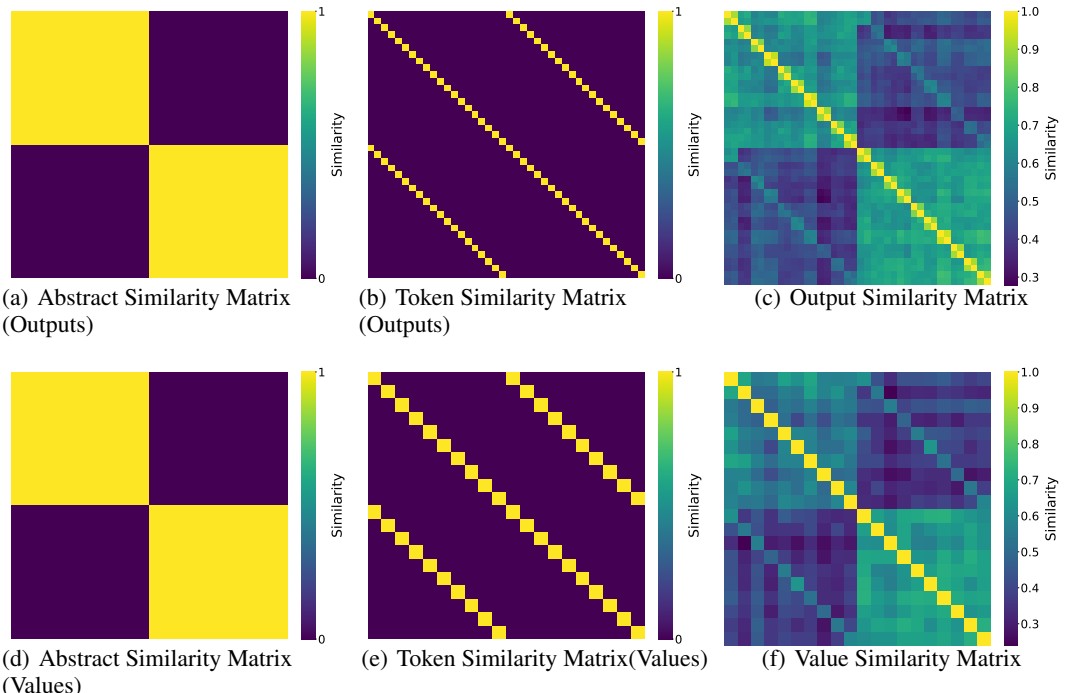

(a) Abstract Similarity Matrix (Outputs)

(b) Token Similarity Matrix (Outputs)

(c) Output Similarity Matrix

(d) Abstract Similarity Matrix (Values)

(e) Token Similarity Matrix(Values)

(f) Value Similarity Matrix

*Figure 13.* **Representational Similarity Analysis for Symbolic Induction Heads (Outputs and Values).** (a), (b), (d), and (e) are the predicted abstract or token similarity matrices; (c) and (f) are the actual embedding similarity matrices.

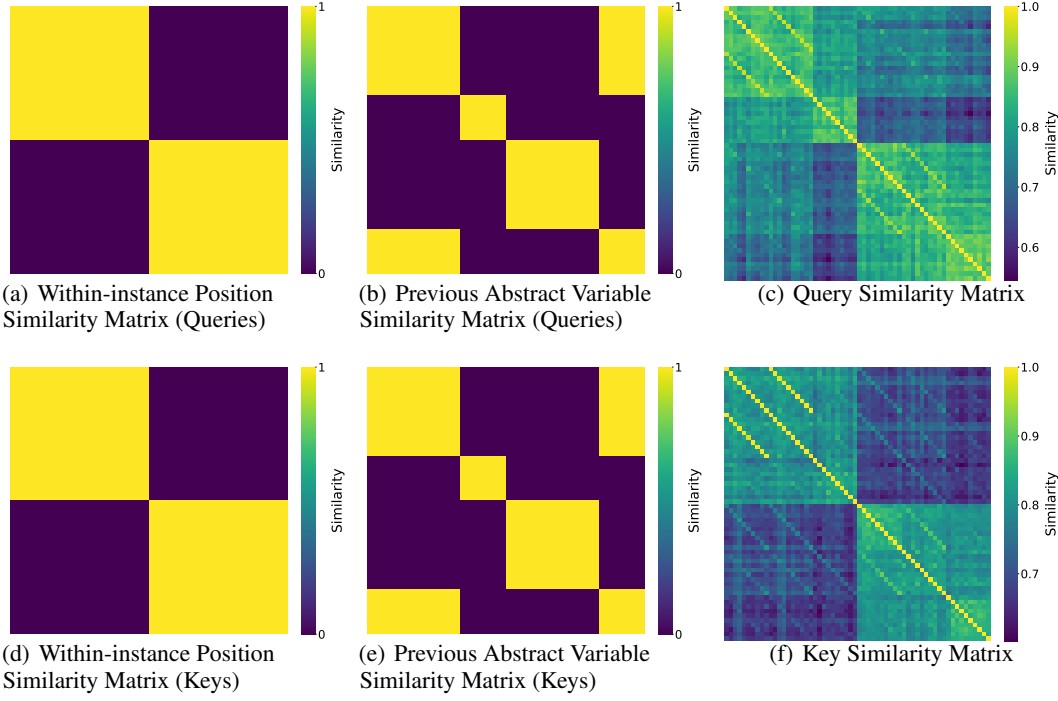

(a) Within-instance Position Similarity Matrix (Queries)

(b) Previous Abstract Variable Similarity Matrix (Queries)

(c) Query Similarity Matrix

(d) Within-instance Position Similarity Matrix (Keys)

(e) Previous Abstract Variable Similarity Matrix (Keys)

(f) Key Similarity Matrix

*Figure 14.* **Representational Similarity Analysis for Symbolic Induction Heads (Queries and Keys).** (a), (b), (d), and (e) are the predicted abstract or token similarity matrices; (c) and (f) are the actual embedding similarity matrices.

The Pearson correlation coefficients for all representational similarity analyses (representing the correlation between each hypothesized similarity matrix and the similarity matrix for a specific attention head component) are shown in Tables 2 and 3. These coefficients were computed based on the lower triangular parts of each similarity matrix (excluding the diagonal).

| Embedding | Symbol Abstraction Heads | | Symbolic Induction Heads | | Retrieval Heads | |
| --- | --- | --- | --- | --- | --- | --- |
| | Abstract RSA | Token RSA | Abstract RSA | Token RSA | Abstract RSA | Token RSA |
| Queries | 0.30 | 0.41 | — | — | 0.57 | 0.06 |
| Keys | 0.62 | 0.22 | — | — | 0.50 | 0.23 |
| Values | 0.67 | 0.17 | 0.69 | 0.19 | 0.07 | 0.55 |
| Outputs | 0.56 | 0.24 | 0.69 | 0.03 | 0.11 | 0.35 |

*Table 2.* **RSA Summary Results.** Pearson correlation between hypothesized similarity matrices and similarity matrices for specific model components. Each column represents the pearson correlation between a hypothesized similarity matrix (abstract vs. token RSA) and the similarity matrix for a particular component (query, key, value, or output embeddings) of a particular set of attention heads. Output RSA results correspond to those shown in Figure 4. Similarity matrices for query, key, and value embeddings are shown in Figures 11-13. For the key and query embeddings in symbolic induction heads, we tested two additional hypotheses requiring a modified task (see Table 3).

| Embedding | Previous Abstract Variable RSA | Within-instance Position RSA |
| --- | --- | --- |
| Queries | 0.33 | 0.63 |
| Keys | 0.29 | 0.73 |

*Table 3.* **Symbolic Induction Head RSA Results.** We tested two hypotheses for the key and query embeddings in the symbolic induction heads. The first hypothesis (Previous Abstract Variable RSA) was that these embeddings represent the abstract variable at the previous position. This hypothesis is based on the induction head mechanism, where keys and queries represent the previous token. The second hypothesis (Within-instance Position RSA) was that these embeddings represent the relative position of a token within the current in-context example. The results indicated that the key and query embeddings more closely matched the second hypothesis. Similarity matrices are shown in Figure 14.

## C.4. Ablation Results

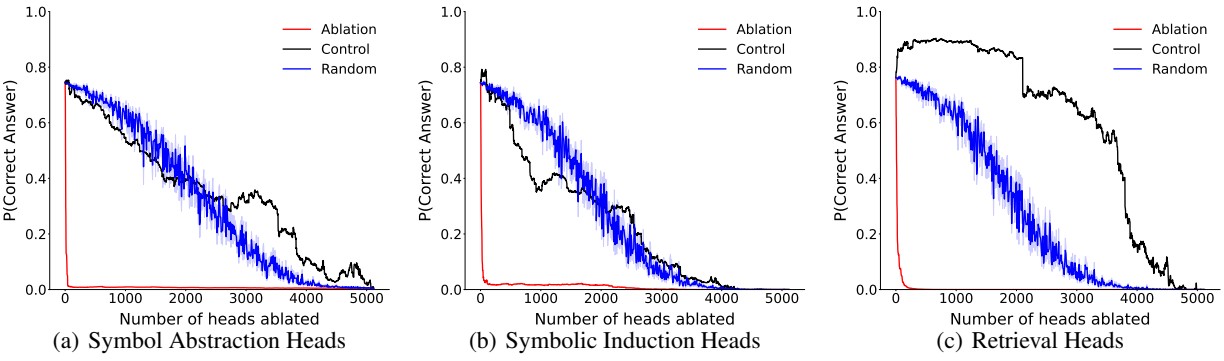

(a) Symbol Abstraction Heads     (b) Symbolic Induction Heads     (c) Retrieval Heads

*Figure 15.* **Ablation Analyses.** **(a)** Cumulative ablation of symbol abstraction heads in decreasing order of causal mediation scores. *Control* condition involves ablation of heads in the same layers but with the lowest causal mediation scores. *Random* refers to randomly selecting heads for ablation. **(b)** Cumulative ablation of symbolic induction heads. **(c)** Cumulative ablation of retrieval heads. Error bars for random baseline reflect standard deviation over 10 runs.

## C.5. Evaluating Other Language Models

Figure 16 summarizes the accuracy of 13 models on identity rule tasks averaged over both ABA and ABB rules and the number of significant heads found in each model. Figure 17 shows the model performance for problems involving either ABA or ABB rules separately. Figures 18-22 show the causal mediation results for this task in all models. Note that GPT-2 does not show robust evidence for the presence of symbol abstraction heads (Figure 19).

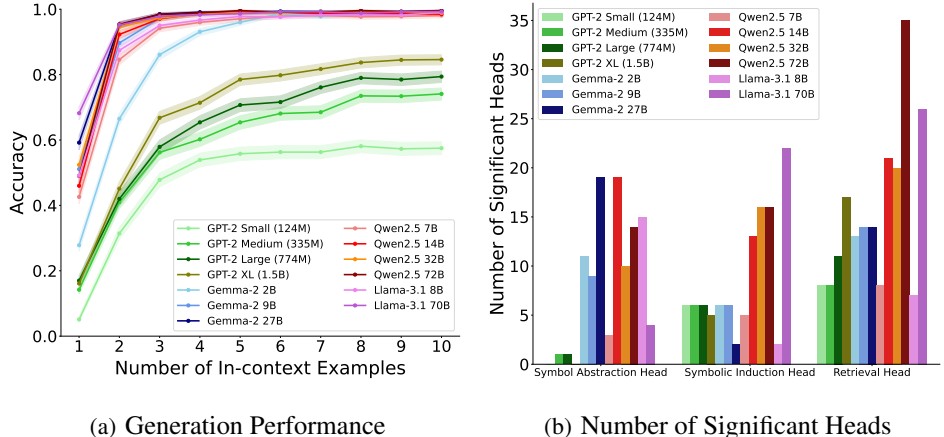

(a) Generation Performance

(b) Number of Significant Heads

*Figure 16.* **Results for different language models on the identity rules task. (a)** Accuracy for each model as a function of the number of in-context examples. Error bars reflect 95% binomial confidence intervals. **(b)** Number of statistically significant symbol abstraction heads, symbolic induction heads, and retrieval heads in each model. We applied the same causal mediation analyses as in Section 3.1 using 10-shot prompts. Permutation testing was performed to estimate the family-wise error rate across all heads, and statistical significance was determined based on a threshold of $p < 0.05$.

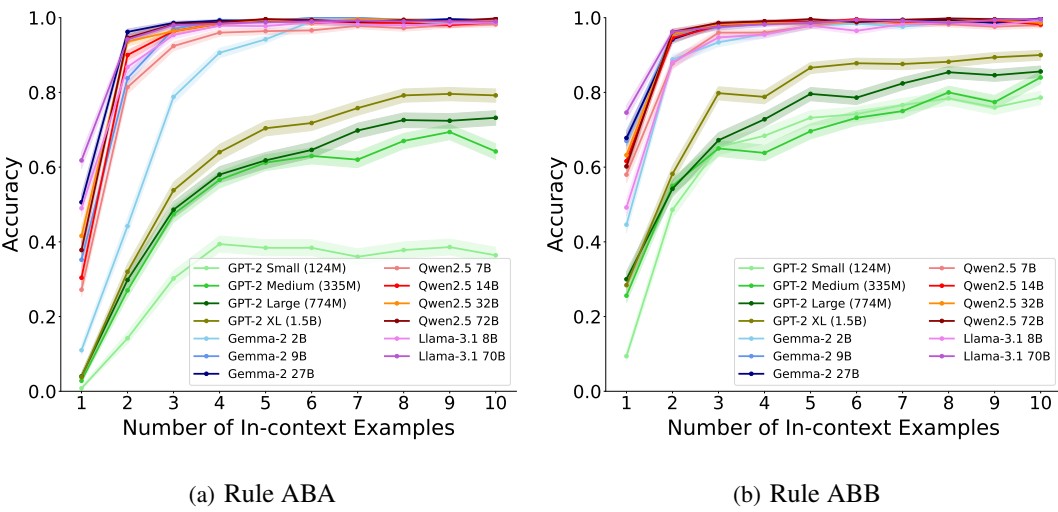

(a) Rule ABA

(b) Rule ABB

*Figure 17.* **Model Performance for Identity Rules Tasks.** We tested model performance on 500 prompts each for the ABA and ABB rules, across different numbers of in-context examples. Error bars reflect 95% binomial confidence intervals.

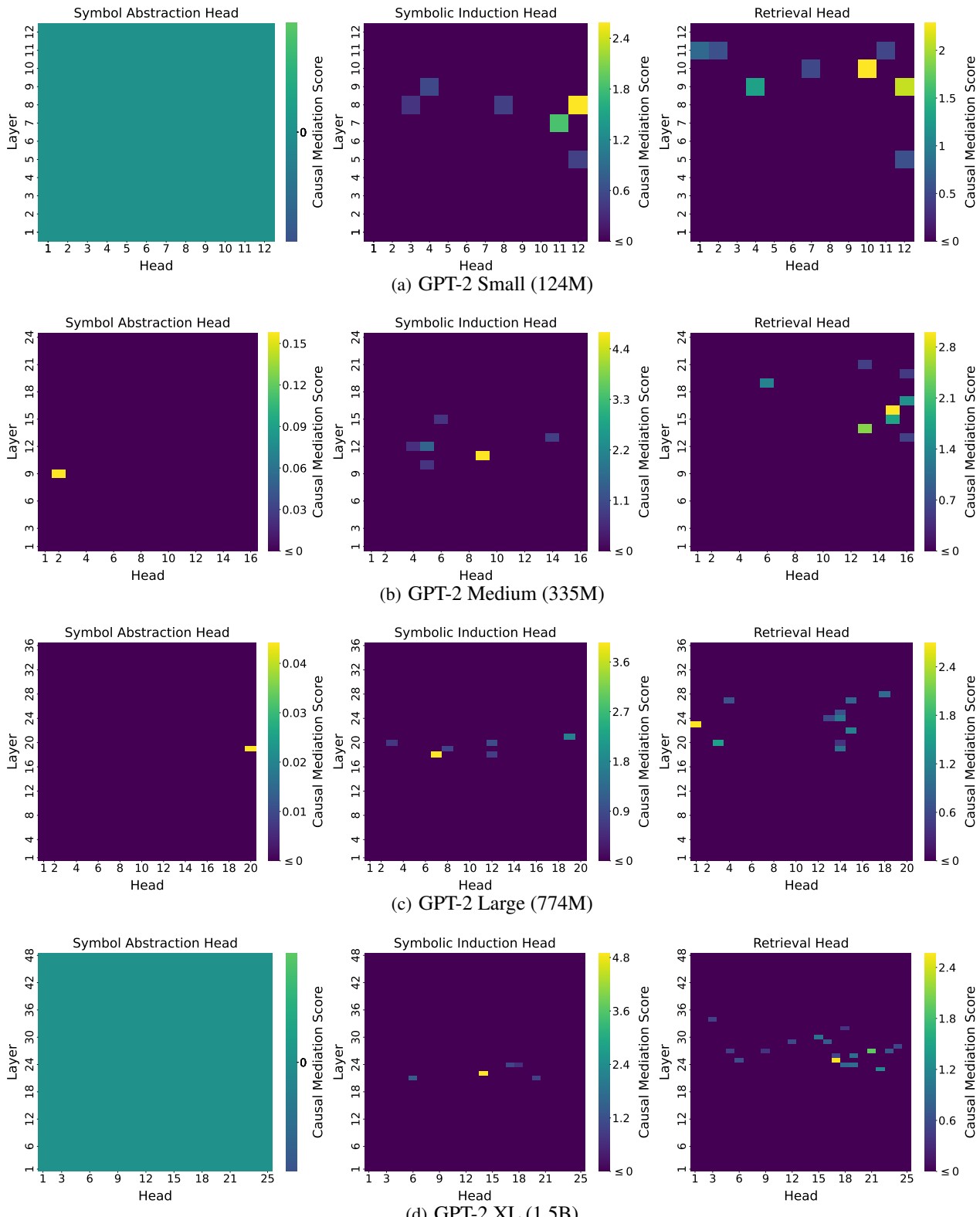

*Figure 18.* **Causal Mediation Results for GPT-2 Models.** From left to right, the heatmaps display significant symbol abstraction heads, symbolic induction heads, and retrieval heads. Permutation testing was performed to estimate the family-wise error rate, and statistical significance was determined based on a threshold of $p < 0.05$.

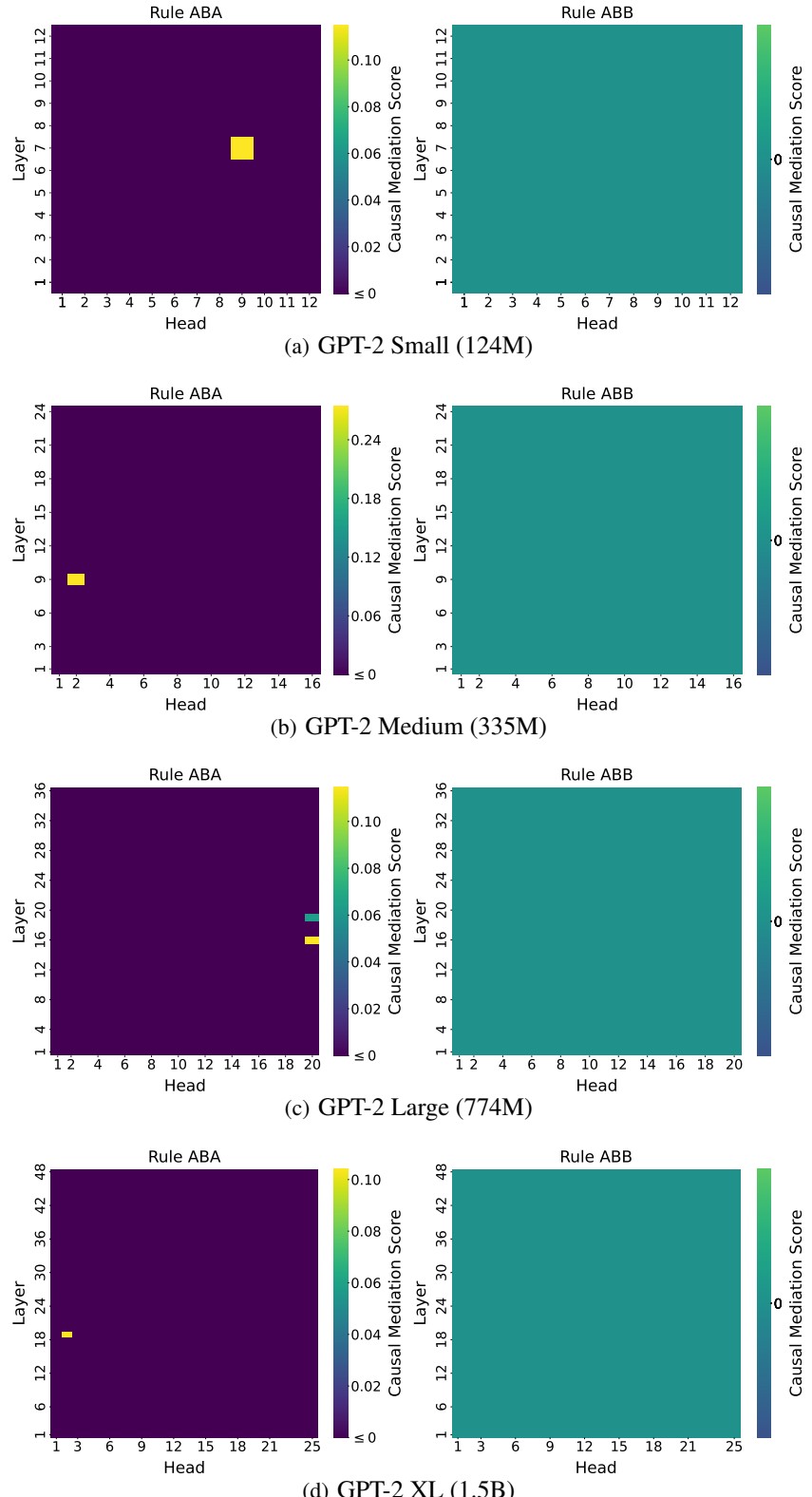

*Figure 19.* **Symbol Abstraction Heads in GPT-2 Models for ABA vs. ABB Rules.** Permutation tests revealed (Left) very few significant abstraction heads for problems involving ABA rules, and (Right) no significant abstraction heads for problems involving ABB rules.

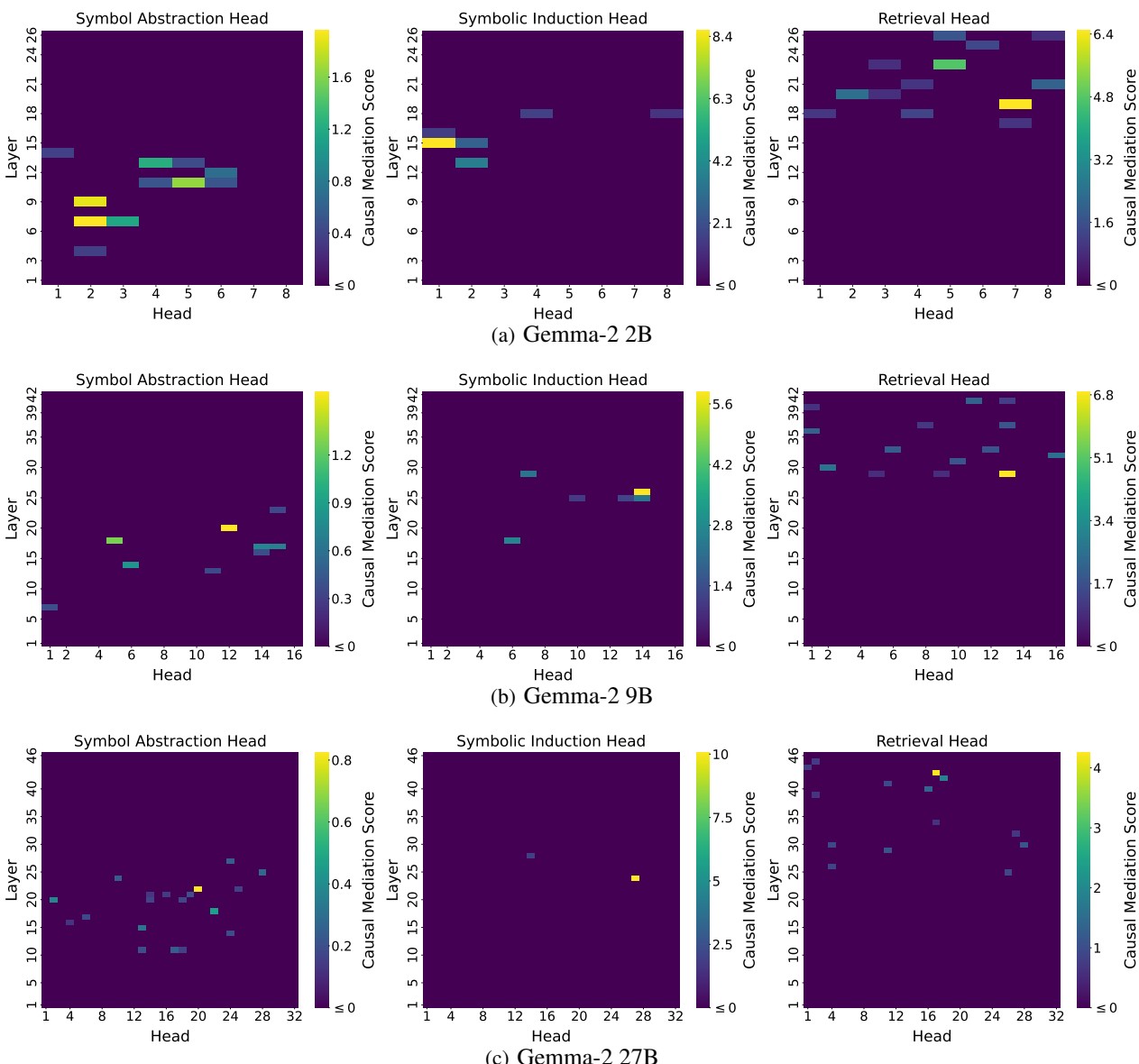

Figure 20. **Causal Mediation Results for Gemma-2 Models.** From left to right, the heatmaps display significant abstraction heads, symbolic induction heads, and retrieval heads. Permutation testing was performed to estimate the family-wise error rate, and statistical significance was determined based on a threshold of $p < 0.05$.

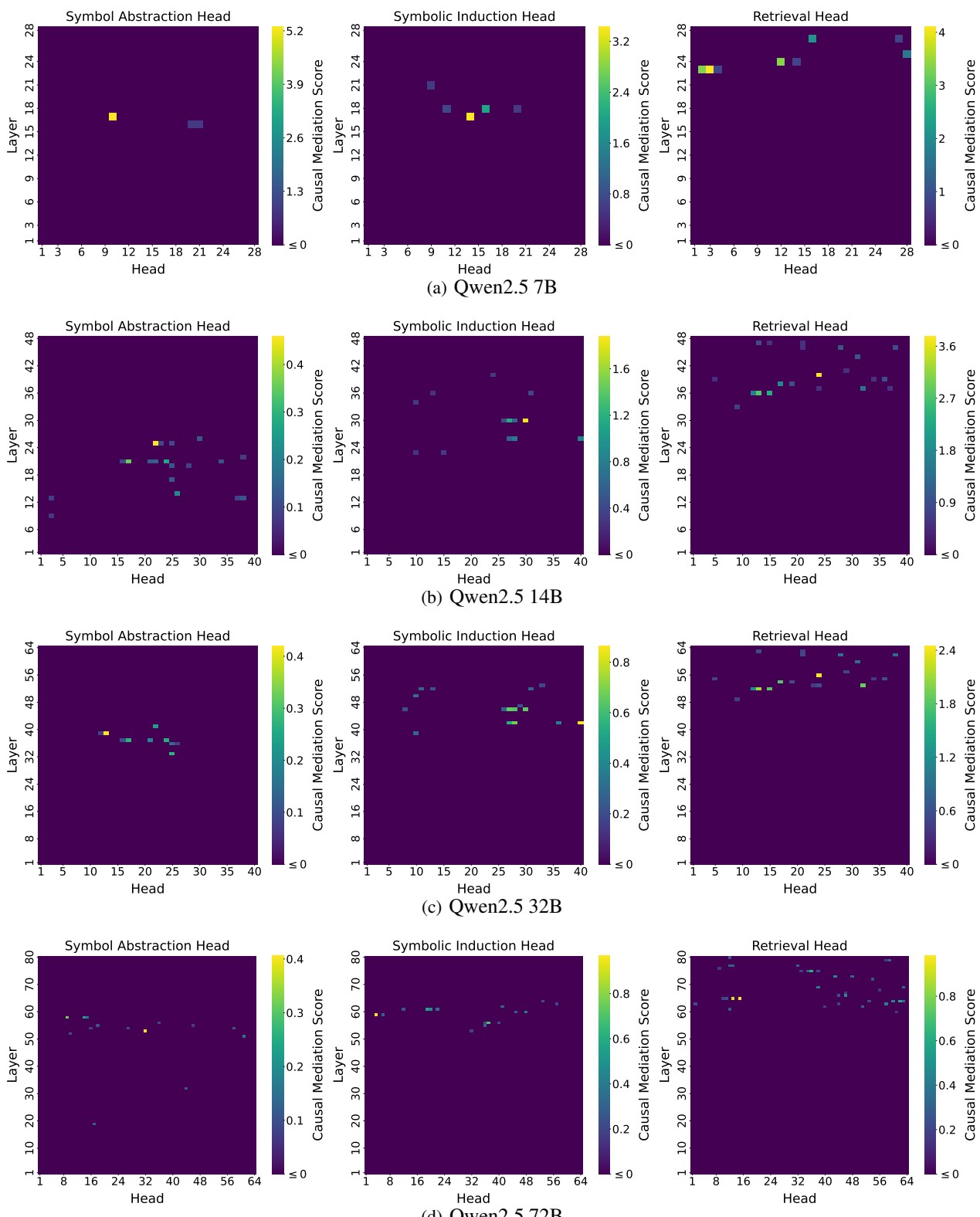

*Figure 21.* **Causal Mediation Results for Qwen2.5 Models.** From left to right, the heatmaps display significant abstraction heads, symbolic induction heads, and retrieval heads. Permutation testing was performed to estimate the family-wise error rate, and statistical significance was determined based on a threshold of $p < 0.05$.

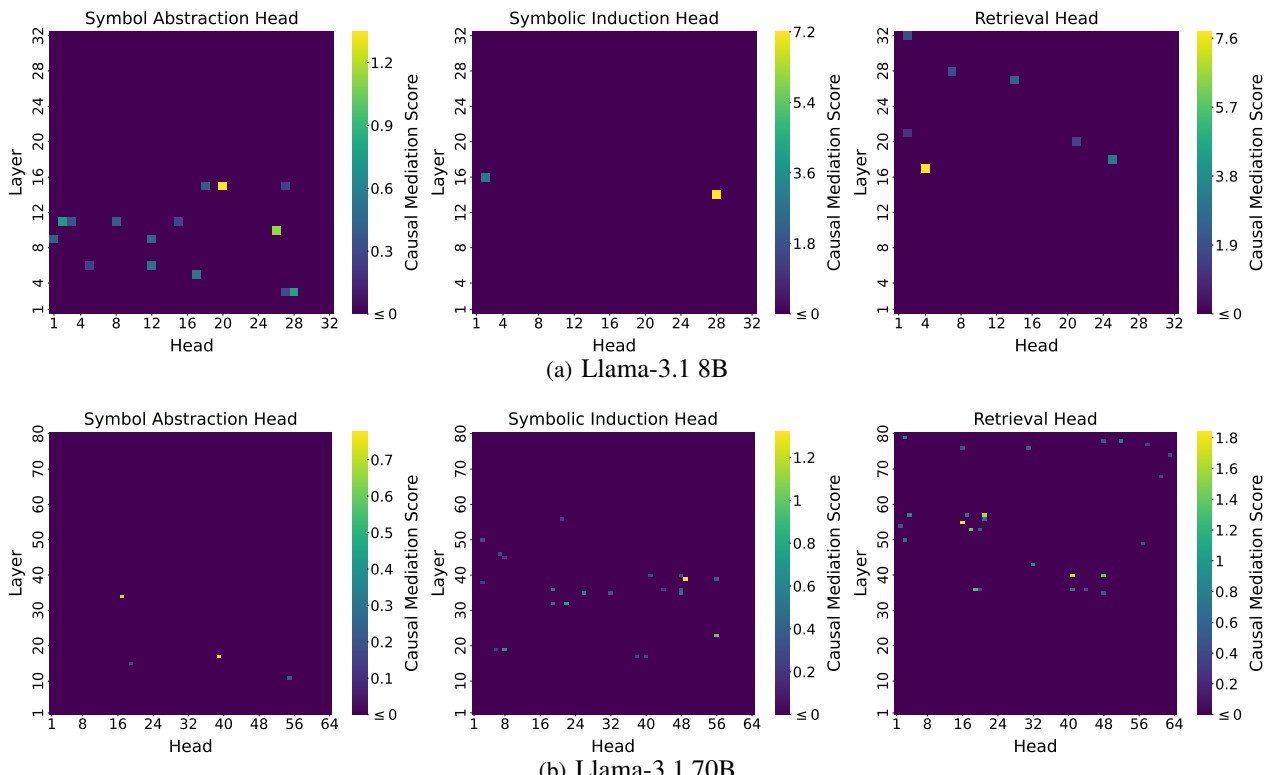

*Figure 22.* **Causal Mediation Results for Llama-3.1 Models.** From left to right, the heatmaps display significant abstraction heads, symbolic induction heads, and retrieval heads. Permutation testing was performed to estimate the family-wise error rate, and statistical significance was determined based on a threshold of $p < 0.05$.

## C.6. More Complex Abstract Reasoning Tasks



Letter string analogies

Successor               Predecessor

[ i j k ] [ i j l ]     [ j k l ] [ i k l ]

[ b c d ] [ b c e ]     [ c d e ] [ b d e ]

[ w x y ] [           [ x y z ] [

Verbal analogies

Synonym                 Antonym

impoverished    :    poor         rich     :     poor

courageous    :    brave        fearful    :    brave

lazy       :             energetic    :



*Figure 23.* **Letter String Analogy and Verbal Analogy Tasks**. Letter string analogy problems involved either successor or predecessor relations. Verbal analogy problems involved either antonym or synonym relations. Similar to the identity rules task, problems were constructed so as to enable the creation of pairs of problems that involved the same completion tokens governed by different abstract rules. For instance, in the example verbal analogies, the answer to both problems is 'idle'. Holding the completion tokens constant, while varying the abstract rule, enabled the causal mediation analysis to dissociate attention heads involved in coding abstract variables vs. literal tokens.

Figure 23 depicts the letter string analogy and verbal analogy tasks. Similar to the design of the identity rules task, these tasks were designed in a manner that allowed a dissociation between representations of literal tokens vs. abstract variables.

C.6.1. LETTER STRING ANALOGY TASK

**Task Setup.** Letter string analogies were originally developed by (Hofstadter et al., 1995). Here, we adopted an implementation similar to (Webb et al., 2023), who developed a letter string analogy dataset for testing LLMs. Our dataset involved 2-shot problems, governed either by a successor or a predecessor relation. For the successor relation, a set of three letters (e.g., [i j k]) was transformed such that the final letter was converted into its alphabetic successor (e.g., [i j *l*]). For the predecessor relation, a set of three letters (e.g., [j k l]) was transformed such that the *first* letter was converted into its alphabetic *predecessor* (e.g., [*i* k l]). This design made it possible to construct pairs of problems using the same set of tokens, where the first and third letters of the transformed set were the same in both the successor and predecessor problems. This was important for our causal mediation analyses, as described below. We created 500 prompts each for the predecessor and successor tasks, randomly sampling token sets for each pair of problems. We evaluated Llama-3.1 70B on this task, counting responses as correct only if all three letters in the completion set were correct. The model achieved $99.2\%$ and $82.0\%$ 2-shot accuracy for the successor and predecessor tasks respectively.

**Causal Mediation Analyses.** We performed causal mediation analysis to identify attention heads involved in representing either abstract variables or literal tokens. By using the same set of tokens to create successor vs. predecessor problems, we were able to dissociate attention heads involved in these two processes. Specifically, the successor and predecessor problems for each pair had completion sets that shared the first and third letter. For instance, given the following instance of a successor relation: [i j k] [i j l], and the following instance of a predecessor relation: [j k l] [i j l], the first and third letters in the completion sets of both problems are 'i' and 'l'. But despite involving the same tokens, these letters shared different relations with the corresponding letters in the first set. Specifically, in the successor relation, the first letter stays the same, and the third letter is transformed into its successor, whereas in the predecessor relation, the first letter is transformed into its predecessor, and the third letter stays the same. To target mechanisms involved in representing abstract variables, we therefore focused our causal mediation analysis on the first and third letters in the completion set.

To target symbol abstraction heads, we patched attention head outputs from corresponding successor to predecessor problems (or vice versa) at the position of the first and third letters in the completion sets for the two in-context examples. For instance, in the example shown in Figure 23, patching was performed on the letters 'i' and 'l' in the second set of brackets on the first row, and the letters 'b' and 'e' in the second set of brackets on the second row.

To target symbolic induction heads, we first appended the correct answer to the prompt. We then targeted the tokens that immediately *preceded* the first and third letters in the completion set for the final (incomplete) query example. These are the token positions at which the model is required to generate the first and third letters for the completion set. For instance, in the example shown in Figure 23, the correct answer '[w x z]' was appended to the successor problem, and the correct answer '[w y z]' was appended to the predecessor problem. Patching was then performed from the open bracket in the second set of brackets in the third row, and the second letter in the appended correct answer completion sets (e.g., 'x' in the successor problem, and 'y' in the predecessor problem).

The expected answer for both of these causal mediation analyses (i.e., the expected output given that the patching was effective) was obtained by applying the alternative rule to each problem. For instance, in the successor problem shown in Figure 23, the expected completion to '[w x y]' was obtained by applying the predecessor relation, yielding '[v x y]'. We used the *sum* of the causal mediation scores defined in Algorithm 1 at the positions preceding the first and third letters in the completion set.

Finally, to target retrieval heads, we created variants of the problems described above in which the query example used completely different tokens. For instance, given the successor problem shown in Figure 23, with the following complete query example: [w x y] [w x z], we created an alternative problem with the following query example: [q r s] [q r t]. The correct answer was appended to both contexts, and patching was performed between two instances with the same rule (either successor or predecessor), at the positions preceding the three letters in the completion set (i.e., beginning with the open bracket, and including the first and second letters). The expected answer for this analysis is obtained simply by appending the tokens from the alternative context in a pair. For instance, when patching from '[q r s] [q r t]' to '[w x y] [w x z]', the expected completion set is '[q r t]'. We measured the causal mediation scores at the positions preceding all three letters in the completion set.

Figure 24 shows the results of this analysis in Llama-3.1 70B. The analysis was performed on a set of 100 prompts for each rule. Only prompts that the model answered correctly were used. Permutation testing was performed to identify attention heads with scores that were significantly greater than 0 ($p < 0.05$). The results were averaged over both patching directions (i.e., successor→predecessor and predecessor→successor). The results were qualitatively similar to the pattern observed for the identity rules tasks. That is, symbol abstraction heads appeared primarily in early layers, symbolic induction heads appeared primarily in intermediate layers, and retrieval heads appeared primarily in later layers.

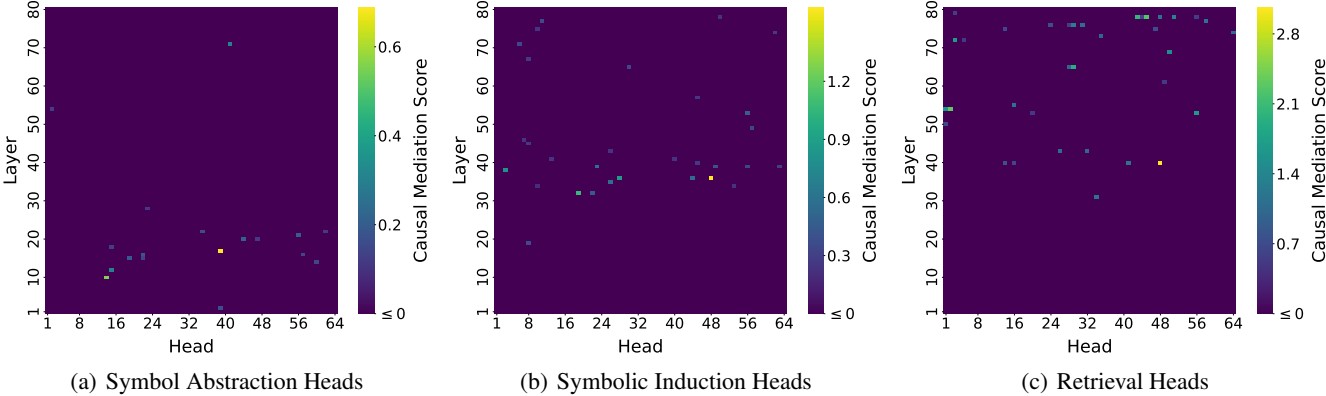

(a) Symbol Abstraction Heads      (b) Symbolic Induction Heads      (c) Retrieval Heads

*Figure 24.* **Letter String Analogy Results.** Causal mediation scores for the letter string analogy task in Llama-3.1 70B. Permutation testing was performed to estimate the family-wise error rate, and statistical significance was determined based on a threshold of $p < 0.05$.

C.6.2. VERBAL ANALOGY TASK

**Task Setup.** We developed a dataset of four-term verbal analogies. Verbal analogies are a common analogy format, involving two pairs of words that share the same relation. We created problems that involved either an antonym or a synonym relation. The format used for presenting problems is shown in Figure 23. We created pairs of problems, one involving an antonym relation, and the other involving a synonym relation, in which the correct answer was the same word. For instance, the pair 'lazy : idle' involves a synonym relation, while the pair 'energetic : idle' involves an antonym relation, despite both pairs ending with the word 'idle'. To create this dataset, we prompted GPT-4 to generate sets of words, in which the first pair of words were synonyms, the second pair of words were also synonyms, but the first and second pairs of words were antonyms with eachother (e.g., lazy, idle, energetic, and active). We then manually inspected each set of words (to ensure that they involved the correct relation), and we filtered any sets that were redundant. This yielded 46 unique word sets.

We then created 500 pairs of synonym and antonym problems by sampling from these word sets. Each problem involved two in-context examples and an incomplete query example, each created from one of the sampled word sets. We then evaluated Llama-3.1 70B on this task. Accuracy was computed by comparing the logit assigned to the correct answer with the logit assigned to an incorrect answer generated by applying the incorrect relation to the query word. For instance, in an antonym problem with the query word 'energetic', the correct answer was 'idle' (generated by correctly applying the antonym relation), and the incorrect answer was 'active' (generated by applying a synonym relation). This procedure ensured that the model was not penalized for generating an unanticipated but correct answer (since there are typically many valid synonyms and antonyms for any given word). Llama-3.1 70B displayed a 2-shot accuracy of $77.0\%$ and $88.4\%$ for the synonym and antonym tasks respectively.

**Causal Mediation Analyses.** To identify attention heads involved in representing abstract variables, we patched from synonym to antonym problems in the same pair, or from antonym to synonym problems. Each pair of problems had the

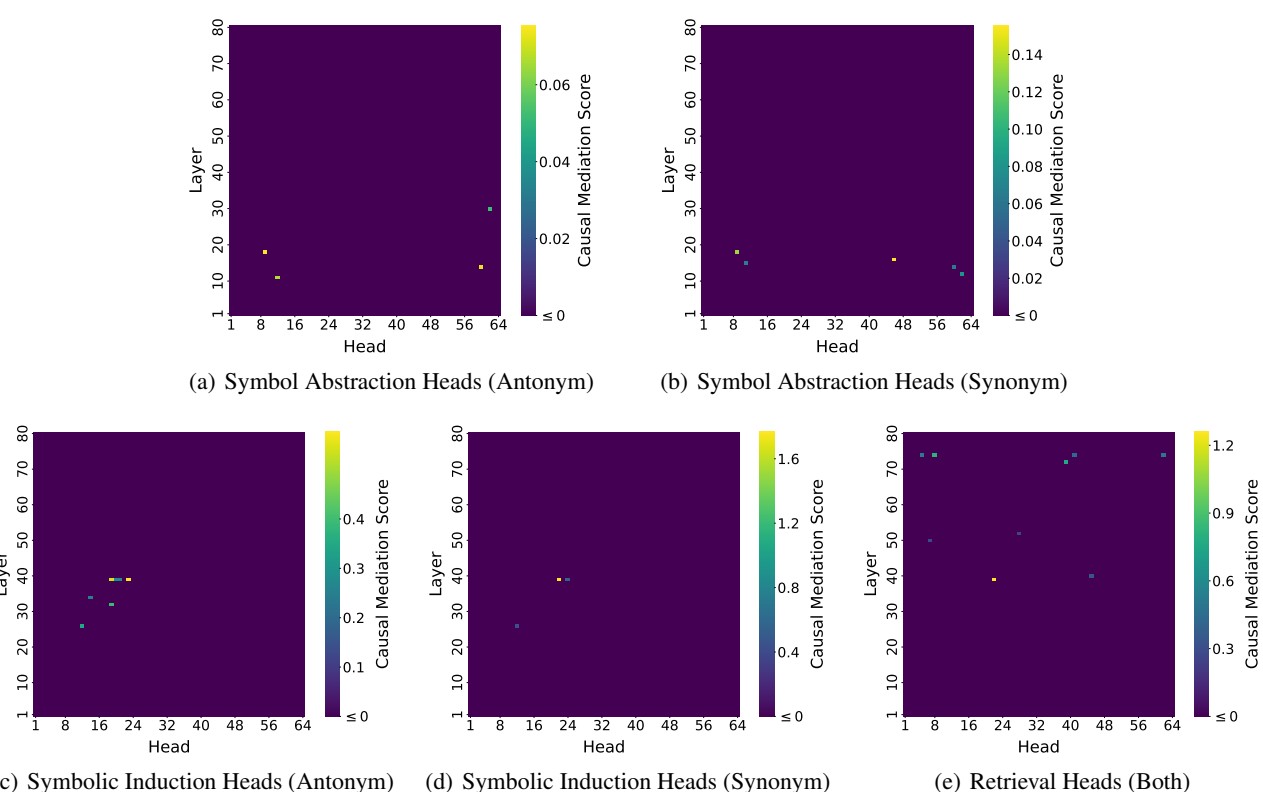

(a) Symbol Abstraction Heads (Antonym)  (b) Symbol Abstraction Heads (Synonym)

(c) Symbolic Induction Heads (Antonym) (d) Symbolic Induction Heads (Synonym) (e) Retrieval Heads (Both)

*Figure 25.* **Verbal Analogy Results.** Causal mediation scores for the verbal analogy task in Llama-3.1 70B. Permutation testing was performed to estimate the family-wise error rate, and statistical significance was determined based on a threshold of $p < 0.05$. Because the scores for symbol abstraction heads and symbolic induction heads have low or even negative within-task correlations, we display separate figures for these heads for antonym vs. synonym relations.

same correct answer, despite involving different relations. To identify symbol abstraction heads, patching was performed at the position of the second word in each of the two in-context examples (e.g., 'poor' and 'brave' in the example shown in Figure 23). To identify symbolic induction heads, patching was performed at the position of the final colon, i.e., the token at which the model was required to generate the completion to the query example. The expected output for both of these analyses was obtained by applying the incorrect relation to the prompt. For instance, when patching from the synonym example to the antonym example in Figure 23, the expected answer (given that patching was effective) was 'active' (generated by applying the synonym relation to 'energetic'). To target retrieval heads, we patched between two problems involving the same relation, but using different sets of words. Patching was again performed at the position of the final colon. We measure the causal mediation scores following Algorithm 1. For answers that involve multiple tokens (i.e., when a single word was tokenized into multiple tokens), instead of computing the causal mediation scores based on logits for individual tokens, we computed these scores based on joint log probabilities for each answer (i.e., summing the log probabilities over all tokens for each answer).

Figure 25 shows the results of this analysis in Llama-3.1 70B. The analysis was performed on a set of 100 prompts for each rule, and only prompts that the model answered correctly were used. The same permutation testing procedure as used in the other tasks was applied. For symbol abstraction and symbolic induction heads, we found that causal mediation scores for the synonym task (identified by patching from synonym to antonym problems) and the antonym task (identified by patching from antonym to synonym problems) were weakly or even negatively correlated (see Table 4). We therefore present these results separately for antonym vs. synonym relations. For retrieval heads, we found that the scores for these relations were correlated, so results were averaged over both patching directions. The results were again consistent with the hypothesized three-stage architecture, with symbol abstraction, symbolic induction, and retrieval heads appearing in early, intermediate, and later layers respectively.

| Attention Head Type | IR | LSA | VA |
|---|---|---|---|
| Symbol Abstraction Heads | 0.49 | 0.33 | 0.19 |
| Symbolic Induction Heads | 0.54 | 0.57 | -0.63 |
| Retrieval Heads | 0.83 | 0.94 | 0.46 |

*Table 4.* **Within-task Correlation Results.** Results reflect Pearson correlation coefficient between causal mediation scores for two relation types within each task. Each column represents a task. IR: identity rules (ABA vs. ABB); LSA: letter string analogy (successor vs. predecessor); VA: verbal analogy (antonym vs. synonym). All correlation coefficients pass two-sided permutation tests ($p < 0.05$).

| Head Score Type | IR vs LSA | VA vs LSA | VA vs IR |
|---|---|---|---|
| Symbol Abstraction Heads | 0.20 | 0.08 | 0.08 |
| Symbolic Induction Heads | 0.42 | 0.34 | 0.36 |
| Retrieval Heads | 0.43 | -0.05 | -0.10 |

*Table 5.* **Between-task Correlation Results.** Results reflect Pearson correlation coefficient between causal mediation scores for two separate tasks. Each column represents a correlation between two tasks. IR: identity rule; LSA: letter string analogy; VA: verbal analogy. All correlation coefficients pass two-sided permutation tests ($p < 0.05$).

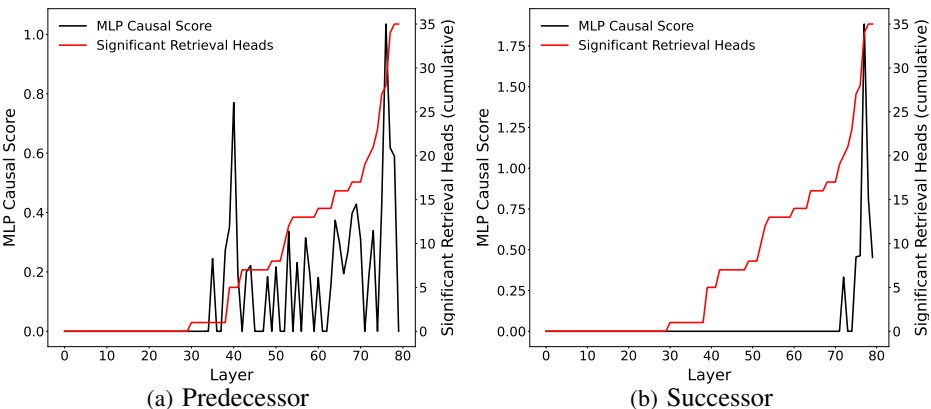

(a) Predecessor (b) Successor

*Figure 26.* **MLP Involvement in Computing More Complex Relations.** For tasks that do not involve same/different relations, relational inference cannot be performed by retrieval alone. We tested the role that the MLP layer played in computing relational transformations for more complex relations. For the letter string analogy task, we found that causal mediation effects in MLP layers were primarily concentrated after the appearance of retrieval heads. This result suggests that the model first retrieves the corresponding letter from the incomplete query example, and then uses subsequent MLP layers to apply the relational transformation (i.e., successor or predecessor).

### C.7. Decoding Abstract Variables

We trained a one-layer linear probe to decode abstract variables (A vs. B) based on the outputs of symbol abstraction and symbolic induction heads. We used the heads identified in Llama-3.1 70B. For symbol abstraction heads, we used outputs corresponding to the position of the final item in each in-context example. For symbolic induction heads, we used outputs corresponding to the final position in the sequence. We created 500 prompts for each rule (ABA and ABB), each involving completely disjoint token sets. We split these into a training set of 200 prompts, a validation set of 100 prompts, and a test set of 200 prompts. Importantly, *the prompts in the training and test sets involved completely disjoint sets of tokens.* This ensured that any ability to generalize from the training to test set would be based on an invariant representation of the abstract variable, rather than the specific tokens used to create the problems. Table 6 shows the results of this analysis.

| | Symbol Abstraction Head | Symbolic Induction Head |
|---|---|---|
| Decoding Test Accuracy | 98.63% | 98.10% |

*Table 6.* **Abstract Symbol Decoding Results.** Results for linear decoder trained to predict abstract variables based on output of symbol abstraction and symbolic induction heads in Llama-3.1 70B. Results show test accuracy. Training and test sets involved completely disjoint token sets.

### C.8. Error Analyses

We compared RSA results for symbol abstraction and symbolic induction heads in correct vs. error trials. Specifically, we computed the Pearson correlation between the similarity matrices for these attention head outputs and the hypothesized similarity matrix based on abstract variables. The results (Table 7) indicated that abstract variables were better represented in the correct trials than in the error trials.

| | Symbol Abstraction Head | Symbolic Induction Head |
|---|---|---|
| Correct Trials | 0.52 | 0.63 |
| Error Trials | 0.47 | 0.49 |

*Table 7.* **Comparisons of Abstract Variable RSA between Correct Trials and Error Trials.** Scores reflect the correlation between pairwise similarity matrix of head outputs and expected similarity pattern defined by abstract variables. These scores are significantly higher in correct vs. error trials for both symbol abstraction heads and symbolic induction heads (permutation test, $p < 0.05$). RSA correlation scores are averaged across all the statistically significant heads for each head type.

