# OpenReview forum: "Emergent Symbolic Mechanisms Support Abstract Reasoning in Large Language Models"
_ICML.cc/2025/Conference — ICML 2025 poster_

### Official Review · Reviewer_Pe4s · 2025-03-09

**Overall Recommendation:** 3

**Summary:**

This paper studies an important topic of whether symbolic representations and behaviors emerge in LLMs when performing abstract reasoning tasks. The paper argues that symbolic abstraction and operations appears at different levels of attention (low-level abstraction, mid-level prediction, high-level retrieval), and verified these hypothesis via causal mediation analysis, attention plots and ablation studies. All analysis was conducted on a simple reasoning tasks (generating strings following the pattern ABA or ABB from two-shot examples), and demonstrated that the attention analysis aligns well with the expected behavior under these controlled experiments.

**Claims And Evidence:**

In a narrow sense, the evidence provided by the paper, as given by the experimental section, supports well the paper's claim that LLMs demonstrated remarkable symbolic behavior for the "ABA, ABB" reasoning task.
However, to claim that LLMs developed intrinsic symbolic mechanism for general abstract reasoning tasks (as the title and introduction implies), more evidences and analysis need to be provided beyond the simple "ABA, ABB" task.

**Essential References Not Discussed:**

The paper could have cited a highly-related recent paper:
Emergent Symbol-like Number Variables in Artificial Neural Networks, Satchel Grant, Noah D. Goodman, James L. McClelland, arXiv 2025

**Experimental Designs Or Analyses:**

The experiments are well-designed and analyzed from the perspective of causal intervention.

**Methods And Evaluation Criteria:**

The methods (causal mediation analysis, attention ablation etc.) makes a lot of sense for the purpose of identifying symbolic behaviors, under the controlled setup of performing "ABA, ABB" tasks.

**Other Comments Or Suggestions:**

In Sec 3.1, right above Eq. (1), it should be "that instantiated an BAA rule" instead of "...that instantiated an ABB rule"?

**Other Strengths And Weaknesses:**

Strength:
The paper studies a really important and intriguing topic of the emergence and role of symbolism in neural networks. If it is successfully proven that symbolic and formal logic abstractions do exist in LLMs internal representations as an emergent phenomenon, this would significantly help researchers understand the inner-working of LLMs, and even how intelligence works in general.

Weakness:
The cases studied by the paper is purely synthetic and a little over-simplistic. They are not strong enough to support the general claim that symbolic abstraction emerges from LLM reasoning process.

**Questions For Authors:**

1. The specific examples ("ABA", "ABB") studied by the paper are highly structured and position-driven. While it is plausible that the model has developed an abstraction of the symbolic patterns from the in-context examples, these specific tasks are perhaps too simple for the paper to draw reliable conclusion that symbolic mechanisms are emergent from LLM reasoning process.

To systematically study the symbolic behaviors of LLM, it would be more convincing to conduct experiments on general reasoning and logical inference tasks, for example maths and arithmetics, first-order logic etc.

2. The study of the paper is based on few-shot learning setup. It remains unclear whether the symbolic abstraction behavior still holds in broader LM use cases like instruction following. For example, instead of giving concrete examples for ABA or ABB, just instruct the model with "Generate 10 random triplets, each following the pattern ABB". Will we still be able to observe similar abstraction mechanism as the paper analyzed?

3, Given that modern LLMs have already demonstrated impressive capability in formal reasoning tasks, it is not hard to believe that LLMs already developed intrinsic symbolic representations and inference rules for these tasks. However, one missing piece is to understand how LLMs developed such a capability from training data. It would be great if the paper provides some analysis or just intuition on how the LLMs acquired such capability, and what types of training data is critically responsible for the symbolic emergence.

4. The paper argues that "we hypothesize that the value embeddings in these heads do not represent the identity of the input tokens, but instead represent only their position." (line 132). If this is the case, then the value embeddings is decoupled from semantic representation, and simply became a position indicator. Wouldn't it be easier to verify this hypothesis just by clustering value embeddings from different examples? If the embedding contains little semantic info, then these vector clusters should be clearly position-dependent.

5. It was not very clear how the retrieval heads work: If like what the paper argues (the retrieval head retrieves actual tokens closest to the latent symbolic variable), then shouldn't Fig. 4 (e) be a plot between variables (A, B) and the token vocabulary?

**Relation To Broader Scientific Literature:**

This paper is well-contextualized connected to the established literature on the philosophical symbolism of artificial intelligence, as well as the historical development of neurosymbolic integration in AI model design.

**Theoretical Claims:**

The paper could have provided a deeper theoretical foundation on the connection between connectionism and symbolism, as well as elaborated on why symbolism has been historically essential in the development of AI.

---

> ### Author Rebuttal · Authors · 2025-04-01
>
> Thank you very much for the thoughtful and detailed feedback. We present detailed responses below to address each of the issues raised. Throughout these responses, we refer to new results that can be viewed here:
>
> https://anonymous.4open.science/r/RB-F30A/13386.pdf
>
> ## Additional models and tasks
> We have tested **12 additional models** (Figures 1, 6, 8-10), including GPT-2 (small, medium, large, and extra large), Gemma-2 (2B, 9B, and 27B), QWEN-2.5 (7B, 14B, 32B, and 72B), and Llama-3.1 8B (along with our original tests on Llama-3.1 70B), and **two additional tasks** (Figures 2-4) including a letter string analogy task and a verbal analogy task. With the exception of GPT-2, we find that our results are qualitatively replicated across all of these models and tasks. These results strongly suggest that the identified mechanisms are a ubiquitous feature of abstract reasoning in sufficiently large language models.
>
> ## Direct analysis of key, query, and value embeddings
> To gain a more precise understanding of the identified attention heads, we have now performed additional analyses applying RSA to the key, query, and value embeddings (Tables 1-2 and Figures 11-14). For abstraction heads, we found that queries primarily represented token identity, keys represented a mixture of both tokens and abstract variables, and values primarily represented the abstract variables. For symbolic induction heads, we found that queries and keys primarily represented the relative position within each in-context example, while values primarily represented abstract variables. For retrieval heads, we found that queries primarily represented abstract variables, keys represented a mixture of both tokens and variables, and values primarily represented the predicted token. These results further confirm the hypothesized mechanisms, namely that abstraction heads convert tokens to variables, symbolic induction heads make predictions over these variables, and retrieval heads convert symbols back to tokens.
>
> ## Clarification about retrieval heads
> The hypothesis about retrieval heads is not that they retrieve tokens that are most *similar* to the abstract variable. The hypothesis is that they retrieve the tokens that are *bound* to the abstract variable. For example, for an ABA rule, with the following incomplete query example:
>
> la (A), li (B), ?
>
> The final variable is A, and in this example A is bound to the token ‘la’. Retrieval heads take the predicted variable (A) as input, and retrieve the token that’s bound to it (‘la’) to predict the token that will come next. To make this clearer in the revised paper, we will clearly describe this in terms of variable-binding, rather than using the ambiguous phrase ‘associated with’.
>
> ## Intuitions on the origins of emergent symbolic mechanisms
> We agree that it is interesting to consider the factors leading to the emergence of symbolic mechanisms. In the discussion section, we have already included some discussion of aspects of the transformer architecture that may contribute (innate similarity mechanisms, indirection), but aspects of the training data may also contribute, including: 1) the inherently relational nature of language, 2) factors such as ‘burstiness’ that have been related to in-context learning [1], and 3) the massive scale and generality of LLM training data, which may force the development of more general-purpose mechanisms. Directly analyzing the relative contribution of these factors will necessitate training language models from scratch under various conditions, which would require extensive resources, but we consider this an important avenue for future work. We will add further discussion of these issues to the revised paper.
>
> [1] Chan, S., Santoro, A., Lampinen, A., Wang, J., Singh, A., Richemond, P., ... & Hill, F. (2022). Data distributional properties drive emergent in-context learning in transformers. Advances in neural information processing systems, 35, 18878-18891.
>
> ## Additional related work
> Thank you for bringing the paper from Grant et al. to our attention. We agree that it is highly relevant. Relative to our work, their study investigates smaller neural networks trained from scratch on specific tasks rather than large pre-trained models, and also investigates a different task setting, but the high-level emphasis on emergent symbolic representations is closely related to our work, and we will make sure to cite it in our revised paper.
>
> ## Clarification on description of 2
> We will correct the description of equation 2 to describe it as a ‘BAA’ rule, while also clarifying that ABB and BAA rules are equivalent in this task (they involve the same pattern of relations).

---

### Official Review · Reviewer_yD8H · 2025-03-13

**Overall Recommendation:** 4

**Summary:**

This paper investigates the mechanisms behind how Large Language Models (LLMs) perform two simple abstract reasoning tasks related to algebraic identity rules (left and right). They identify three types of attention heads: abstraction heads, symbolic induction heads, and retrieval heads, which are implicated in performing the task. The attention patterns match the predicted values very closely, and show evidence that LLMs do indeed possess the capability to do symbolic processing to some extent.

**Claims And Evidence:**

While the setup and experiments are simple, they are thorough and well thought out. I found the claims well-supported by the evidence.

**Essential References Not Discussed:**

The literature awareness of this paper is great! Here are a couple works that might help contextualize the contributions of the paper such as [2] who study what algorithms transformers trained on symbolic data settings such as algebraic groups learn, or [3] which study induction heads for "symbolized" language reasoning, though I think their may still fit the standard token-induction and not symbolic induction studied here. However, I think the work already cited is pretty well-covered.

**Experimental Designs Or Analyses:**

The experiments are well-designed and nice counterfactuals are used to elicit different responses from the network to study information flow/processing.

**Methods And Evaluation Criteria:**

There are no benchmarks used here, just a simple dataset. But the proof-of-concept of symbolic processing is sufficient for the goal of the paper.

**Other Comments Or Suggestions:**

- In Line 155-157, can you clarify why you say the final token is the final token in each in context example, is the same for both contexts? Did you mean that A_N is the target token for both examples?

Typos:
Line 320: patten -> pattern

**Other Strengths And Weaknesses:**

Overall, I think this is a solid paper and gained some new insights while reading it.

I think the experiments section is a little light and I'd like to see additional evidence or more concrete analysis on the connection of the three kinds of attention heads - whether they are directly connected or there are other mediators between them.

- There are only two tasks for which symbolic induction is tested (ABA, and ABB). One thing that could strengthen the claims of this paper is to examine other setups where symbolic induction may be helpful/used. A simple extension that I imagine wouldn't change the results you find are things like ABCABC..., or ABBCABBC...?

**Questions For Authors:**

1. Have you studied these classes of attention heads on a broader class of relations?
2. Are the function vector heads only symbolic induction heads or did they relate to the abstraction heads or retrieval heads you find?
3. Are the abstraction heads you identify a more general form of "previous token heads" identified in previous circuit work [4,5]?


___
[4] Elhage, et al. In-context Learning and Induction Heads. 2022. (https://transformer-circuits.pub/2022/in-context-learning-and-induction-heads/index.html)

[5] Wang, et al. Interpretability in the Wild: a Circuit for Indirect Object Identification in GPT-2 small. (https://arxiv.org/abs/2211.00593)

**Relation To Broader Scientific Literature:**

This work attempts to show that LLMs trained on natural text corpora are capable of developing symbolic processing mechanisms that reason over abstract concepts. I am not aware of previous work that has studied this specifically, though the original induction heads thread from Anthropic did suggest that induction heads may be able to process both literal and abstract relationships. It is nice to see this in a concrete, though simple form.

I will also point out (though do not expect the authors to have been aware of/cited it due to its release date) a recent paper that attempts to clarify the difference between induction heads and function vector heads [1], showing that literal copying and few-shot ICL performance are mediated by different sets of heads.

___

[1] Yin, et al. Which Attention Heads Matter for In-Context Learning? 2025. (https://arxiv.org/abs/2502.14010)

[2] Liu, et al. Transformers Learn Shortcuts to Automata. ICLR 2022. (https://openreview.net/forum?id=De4FYqjFueZ)

[3] Song, et al. Out-of-distribution generalization via composition: a lens through induction heads in Transformers. (https://arxiv.org/pdf/2408.09503)

**Theoretical Claims:**

There are no theoretical claims in the paper.

---

> ### Author Rebuttal · Authors · 2025-04-01
>
> Thank you very much for the thoughtful and detailed feedback. We present detailed responses below to address each of the issues raised. Throughout these responses, we refer to new results that can be viewed here:
>
> https://anonymous.4open.science/r/RB-F30A/13386.pdf
>
> ## Additional models and tasks
> We have tested **12 additional models** (Figures 1, 6, 8-10), including GPT-2 (small, medium, large, and extra large), Gemma-2 (2B, 9B, and 27B), QWEN-2.5 (7B, 14B, 32B, and 72B), and Llama-3.1 8B (along with our original tests on Llama-3.1 70B), and **two additional tasks** (Figures 2-4) including a letter string analogy task and a verbal analogy task. With the exception of GPT-2, we find that our results are qualitatively replicated across all of these models and tasks. These results strongly suggest that the identified mechanisms are a ubiquitous feature of abstract reasoning in sufficiently large language models.
>
> ## Direct analysis of key, query, and value embeddings
> To gain a more precise understanding of the identified attention heads, we have now performed additional analyses applying RSA to the key, query, and value embeddings (Tables 1-2 and Figures 11-14). For abstraction heads, we found that queries primarily represented token identity, keys represented a mixture of both tokens and abstract variables, and values primarily represented the abstract variables. For symbolic induction heads, we found that queries and keys primarily represented the relative position within each in-context example, while values primarily represented abstract variables. For retrieval heads, we found that queries primarily represented abstract variables, keys represented a mixture of both tokens and variables, and values primarily represented the predicted token. These results further confirm the hypothesized mechanisms, namely that abstraction heads convert tokens to variables, symbolic induction heads make predictions over these variables, and retrieval heads convert symbols back to tokens.
>
> ## Additional analyses on relationship to function vectors
> In the original submission, we showed that function vector scores and symbolic induction head scores are very highly correlated, indicating that symbolic induction heads are responsible for producing function vectors. We have now performed additional analyses that suggest a more complex interpretation (Table 4). Specifically, when function vector scores are computed based on the final position in the sequence, these scores are highly correlated with symbolic induction heads, but not with abstraction heads. In contrast, when function vector scores are computed based on the final item in each context-example, these scores are highly correlated with abstraction heads, but not symbolic induction heads. These results suggest that function vectors are first computed by abstraction heads at the level of individual in-context examples, and symbolic induction heads are primarily responsible for aggregating them across in-context examples.
>
> ## Related work on relationship between induction heads and function vectors
> Thank you for bringing the paper from Yin et al. to our attention. We agree it is highly relevant and will make sure to cite it in our revised paper. It is especially interesting that Yin et al. find evidence that function vector heads evolve from induction heads. Our results suggest an explanation for this finding: function vector heads (i.e. symbolic induction heads) can be viewed as performing induction over symbolic (rather than literal) inputs.
>
> ## Additional related work
> Thank you for pointing out the additional related work from Liu et al. and Song et al. We agree that these studies are also relevant, and will make sure to cite them in the revised paper.
>
> ## Relationship to previous-token heads
> It is unlikely that the abstraction heads are performing a function similar to previous-token heads. Previous-token heads copy the token from the position t-1 into the residual stream at position t. By contrast, abstraction heads are responsible for computing *which* token (t-1 or t-2) is related to the token at position t, and what that relationship is. That is, abstraction heads do not copy the token from positions t-1 or t-2. They compute whether the token at position t is related to the token at position t-1 or t-2, and what that relationship is.

---

> > ### Comment · Reviewer_yD8H · 2025-04-02
> >
> > Thank you for the additional experiments for more models and tasks, as well as the clarification - they are helpful and have definitely strengthened the paper! I plan on retaining my score to accept.

---

### Official Review · Reviewer_8kkE · 2025-03-14

**Overall Recommendation:** 4

**Summary:**

The paper studies the internal mechanisms of a Llama3-70B on an in-context learning task. Specifically, they study an abstract reasoning task in which the model is given multiple demonstrations of the form ABA or ABB, where A and B correspond to randomly selected tokens, and on the final example the model has to predict either A or B.

The authors identify a combination of multiple attention heads that are causally responsible for solving the task: (1) *Abstraction heads* extract relational information about the input tokens in each demonstration, (2) *Symbolic induction heads* perform induction over the relational information, (3) *Retrieval heads* predict the subsequent token by retrieving the value associated with the extracted relational variable.

They causally verify their observations using activation patching experiments. Specifically, they verify the existence of token-independent, abstract variables containing relational information. Additionally, they analyse attention patterns and compare the similarity of attention head outputs to evaluate whether they match their expectation.

Finally, they contrast induction heads as identified in Olsson et al. (2022) with the symbolic induction heads identified in their setting. Interestingly, they find that for symbolic attention heads the prefix matching score does not correlate with the causal mediation score, suggesting that these are different from induction heads. Instead, those heads appear to be involved in the creation of function vectors (Todd et al., 2023).

**Claims And Evidence:**

The experiments are generally supported by clear and convincing evidence.

**Essential References Not Discussed:**

There have been a number of papers that mechanistically study some form of reasoning in language models. This includes works such as:
- A. Al-Saeedi and A. Harma, ‘Emergence of symbolic abstraction heads for in-context learning in large language models’, in Proceedings of Bridging Neurons and Symbols for Natural Language Processing and Knowledge Graphs Reasoning @ COLING 2025, 2025, pp. 86–96.
- J. Brinkmann, A. Sheshadri, V. Levoso, P. Swoboda, and C. Bartelt, ‘A Mechanistic Analysis of a Transformer Trained on a Symbolic Multi-Step Reasoning Task’, in Findings of the Association for Computational Linguistics: ACL 2024, 2024, pp. 4082–4102.
- S. Dutta, J. Singh, S. Chakrabarti, and T. Chakraborty, ‘How to think step-by-step: A mechanistic understanding of chain-of-thought reasoning’, Transactions on Machine Learning Research, 2024.
- A. Saparov et al., ‘Transformers Struggle to Learn to Search’, in The Thirteenth International Conference on Learning Representations, 2025.

Given the relevance of these works, it would be valuable to contextualize this study within their findings and highlight how it builds upon or differs from their approaches.

**Experimental Designs Or Analyses:**

I did check the soundness and validity of the experimental design and analyses; most importantly, the activation patching experiments, representational similarity analyses as well as the comparison with induction heads and function vectors.

**Methods And Evaluation Criteria:**

The evaluation criteria, e.g. the causal mediation score for the activation patching experiments, make sense for the problem at hand.

**Other Comments Or Suggestions:**

NA

**Other Strengths And Weaknesses:**

Overall, I think this is solid work aiming at mechanistically understanding in-context reasoning in language models. It builds upon existing work in the field and studies reasoning in an original setting with interesting results and findings. Notably, the authors study this task in Llama3-70B, which presents additional challenges due to its size. Despite this, the paper remains clear and easy to follow.

The main weaknesses are the limited engagement with prior work studying similar settings, and at times a lack of depth in the analyses. For example, it would have been interesting to study failure cases of the model.

**Questions For Authors:**

NA

**Relation To Broader Scientific Literature:**

The paper fits well into the research direction aimed at (mechanistically) understanding reasoning in language models. In contrast to some prior work, this paper studies an in-context reasoning task in which function vectors (Todd et al., 2024) seem to emerge. While Todd et al. (2024) introduce the notion of abstract computations being represented in simple vectors in language models, they do not mechanistically study how any of those emerge. Thus, this paper is an in-depth study of how one specific function vector emerges in a large language model.

**Theoretical Claims:**

The paper does not make any theoretical claims.

---

> ### Author Rebuttal · Authors · 2025-04-01
>
> Thank you very much for the thoughtful and detailed feedback. We present detailed responses below to address each of the issues raised. Throughout these responses, we refer to new results that can be viewed here:
>
> https://anonymous.4open.science/r/RB-F30A/13386.pdf
>
> ## Additional related work
> Thank you for bringing these additional studies to our attention. We agree that they are highly relevant, and will cite them in the revised paper. We include below a discussion of the relationship to our results:
> - Al-Saeedi and Harma identify a type of attention head that they refer to as ‘symbolic abstraction heads’ in smaller transformer models trained from scratch on identity rule tasks. Based on the attention patterns that they present, these heads seem to correspond to what we refer to as ‘symbolic induction heads’. Relative to that work, our study provides many complementary lines of evidence (RSA, attention analyses, ablations, causal mediation), identifies two additional types of heads, and demonstrates the involvement of these mechanisms in large pre-trained language models. However, the setting studied by Al-Saeedi and Harma provides a complementary line of evidence that may be especially useful in future work investigating the contribution of training regime and architecture (which requires training models from scratch).
> - The other studies investigate different types of reasoning, namely logical reasoning (Dutta et al.) and planning (Brinkmann et al. and Saparov et al.). One difference with our study is that language models have generally not been found to be very good at performing these types of reasoning reliably, whereas the analogical reasoning and rule induction setting that we investigate in our work is one where language models have shown stronger performance. This provides an opportunity to understand how language models are solving the types of problems that they seem to be able to solve in a relatively robust manner.
>
> We will add further discussion of these issues to the revised paper.
>
> ## Analysis of failure cases
> We agree that it is important to understand how the identified mechanisms contribute to the model’s failures as well as successes. To address this, we looked at how the RSA results differed between correct and error trials. We found that the outputs of abstraction heads and symbolic induction heads represented abstract variables more precisely (had a higher correlation with the abstract RSA matrix) in correct vs. error trials (Table 5). However, the effect size of this difference was very small for abstraction heads, suggesting that the effect was driven primarily by differences in the symbolic induction heads. One interpretation of these results is that abstraction heads correctly encode symbols on error trials, but symbolic induction heads do not successfully aggregate these results across in-context examples, perhaps due to interference from other heads. We will include these results in the revised paper.
>
> ## Additional results
> We have also included a number of additional results in the rebuttal file, including results for **12 additional models** (GPT-2 small, medium, large, and extra large; Gemma-2 2B, 9B, and 27B; QWEN-2.5 7B, 14B, 32B, and 72B; and Llama-3.1 8B), **2 additional tasks** (letter string analogies and verbal analogies), more detailed analyses of the identified attention heads (looking at the representations formed by keys, queries and values), and further tests of symbolic invariance.

---

### Official Review · Reviewer_ekki · 2025-03-17

**Overall Recommendation:** 3

**Summary:**

This paper investigates the internal mechanisms that support abstract reasoning in LLMs, focusing on the open-source model Llama3-70B. The paper makes a contribution to the ongoing debate about the reasoning capabilities of LLMs by proposing a novel three-stage symbolic architecture and providing empirical evidence to support its existence.
The authors test this hypothesis using a simple but paradigmatic algebraic rule induction task (ABA/ABB rules), where the model must predict the next token in a sequence based on in-context examples. Llama3-70B achieves 95% accuracy on this task, suggesting robust performance. Through a series of mechanistic interpretability techniques—including causal mediation analysis, attention pattern analysis, representational similarity analysis, and ablation studies—the authors identify and validate the roles of the proposed attention heads.

**Claims And Evidence:**

1. The paper claims that the identified mechanisms perform symbolic processing, but the evidence is not entirely convincing. The representations produced by abstraction and symbolic induction heads are not perfectly abstract (they retain some token-specific information), suggesting that the mechanisms may not be fully symbolic.

2. The paper does not provide a clear definition of what constitutes "symbolic" processing in the context of neural networks, which weakens the claim.

3. The study is limited to a single task (ABA/ABB rule induction) and a single model (Llama3-70B). While the task is paradigmatic for studying relational abstraction, it is unclear whether the identified mechanisms generalize to more complex reasoning tasks or other models.

4. The paper does not explore whether these mechanisms emerge in smaller models or models with different architectures, which would strengthen the claim that they are a general feature of LLMs.

5 The claim that the findings resolve the symbolic vs. neural network debate is overstated. The paper does not provide sufficient evidence to conclude that the identified mechanisms are truly symbolic or that they generalize to other tasks and models.

**Essential References Not Discussed:**

N/A

**Experimental Designs Or Analyses:**

The causal mediation analysis is a rigorous and appropriate method for identifying the causal role of specific components (e.g., attention heads) in the model’s behavior. By patching activations from one context to another, the authors isolate the contributions of different heads to the model’s predictions. However, the analysis is somewhat limited to the specific task and model studied, and it does not rule out alternative explanations for the observed behavior (e.g., other emergent mechanisms).
The interpretation of the causal mediation scores depends on the assumptions of the analysis, which are not fully discussed in the paper. For example, it is unclear whether the patching procedure introduces any artifacts or biases.

The experimental designs and analyses in the paper are generally sound and appropriate for the problem and application at hand. The algebraic rule induction task is well-suited for studying abstract reasoning, and the mechanistic interpretability techniques (causal mediation analysis, attention pattern analysis, representational similarity analysis, ablation studies) provide multiple lines of evidence to support the proposed architecture.

**Methods And Evaluation Criteria:**

1. The use of an algebraic rule induction task (ABA/ABB rules) is well-justified for studying abstract reasoning in LLMs. This task requires the model to identify and apply abstract rules (e.g., repetition or alternation) based on in-context examples, making it a paradigmatic case of relational abstraction. The use of arbitrary tokens ensures that the task cannot be solved by relying on statistical patterns, which is critical for isolating abstract reasoning capabilities.

This method has merits: The task is simple yet effective for probing the model’s ability to perform abstract reasoning.
It has been used in prior work to study systematic generalization in neural networks and symbol-processing in human cognition, providing a strong foundation for comparison.

However, The task is relatively simple compared to more complex reasoning tasks, e.g., mathematical reasoning, planning, or analogical reasoning, which might limit the generalizability of the findings.

2. The methods and evaluation criteria used in the paper are generally appropriate for the problem and application at hand.

**Other Comments Or Suggestions:**

1. Could the authors clarify the definition of "symbolic"? The paper frequently uses the term "symbolic" to describe the identified mechanisms, but it is not always clear what this means in the context of neural networks. Providing a clearer definition (e.g., distinguishing between discrete, rule-based symbols and distributed, approximate symbols) would strengthen the paper’s claims and help readers understand the nature of the proposed mechanisms.

2. The paper could expand its discussion of the broader implications of the findings for AI and cognitive science. For example, how might these insights inform the design of more robust and interpretable AI systems? How do these mechanisms compare to human cognitive processes?

3. To strengthen the generalizability of the findings, consider testing the identified mechanisms on a wider range of tasks (e.g., mathematical reasoning, planning, or analogical reasoning) and models (e.g., smaller or larger LLMs, models with different architectures).
Explore whether these mechanisms emerge under different training regimes or datasets, which would provide insights into their dependence on specific training conditions.

4. The paper does not thoroughly rule out alternative explanations for the observed behavior, such as statistical approximations. Including additional experiments or analyses to address these possibilities would make the findings more robust.

5. The paper can also include a more detailed discussion of its limitations, such as the simplicity of the task, the focus on a single model, and the potential dependence on specific architectural features of the transformer. Acknowledging these limitations would provide a more balanced perspective on the findings.

**Other Strengths And Weaknesses:**

Strengths

1 The submission identifies and characterizes new types of attention heads (abstraction heads, symbolic induction heads, retrieval heads) that support abstract reasoning in LLMs. It also provides a detailed, mechanistic account of how these heads work together to perform a form of symbol processing.

2. It contributes to the symbolic vs. neural network debate by demonstrating how neural networks can develop symbol-like processing capabilities without explicit architectural biases. It uses a combination of causal mediation analysis, attention pattern analysis, representational similarity analysis, and ablation studies to validate the proposed mechanisms.

3. It builds on prior work in emergent reasoning, systematic generalization, and mechanistic interpretability, while providing new insights into the role of function vectors and induction heads.

Weakness:

1. The study focuses on a single, relatively simple task (ABA/ABB rule induction) and a single model (Llama3-70B), raising questions about whether the findings generalize to more complex reasoning tasks or other models.

2. The paper claims that the identified mechanisms perform symbolic processing, but the evidence is not entirely conclusive. The representations produced by abstraction and symbolic induction heads are not perfectly abstract (they retain some token-specific information), which weakens the claim.

3 The paper does not thoroughly rule out alternative explanations for the observed behavior, such as statistical approximations.

**Questions For Authors:**

See the above questions and comments.

**Relation To Broader Scientific Literature:**

The contributions of this paper are related to 1. address the debate about the robustness and nature of these capabilities, with some studies questioning whether LLMs rely on structured reasoning or merely approximate it through statistical patterns; 2. mechanistic interpretability of Transformers

**Theoretical Claims:**

No theoretical claims proposed in this paper.

---

> ### Author Rebuttal · Authors · 2025-04-01
>
> Thank you very much for the thoughtful and detailed feedback. We present detailed responses below to address each of the issues raised. Throughout these responses, we refer to new results that can be viewed here:
>
> https://anonymous.4open.science/r/RB-F30A/13386.pdf
>
> ## Additional models and tasks
> We have tested **12 additional models** (Figures 1, 6, 8-10), including GPT-2 (small, medium, large, and extra large), Gemma-2 (2B, 9B, and 27B), QWEN-2.5 (7B, 14B, 32B, and 72B), and Llama-3.1 8B (along with our original tests on Llama-3.1 70B), and **two additional tasks** (Figures 2-4) including a letter string analogy task and a verbal analogy task. With the exception of GPT-2 (see ‘Testing smaller models’ below), we find that our results are qualitatively replicated across all of these models and tasks. These results strongly suggest that the identified mechanisms are a ubiquitous feature of abstract reasoning in sufficiently large language models.
>
> ## Defining ‘symbol processing’
> We define ‘symbol processing’ in terms of two key properties:
> - Symbolic representations are *invariant* to the content of the values that they are bound to. That is, the representation of the abstract variable ‘A’ should be the same regardless of which values this variable is assigned to. Although the abstraction and symbolic induction head outputs preserve some information about specific tokens, we find that they still contain a subspace that represents abstract variables in an invariant manner (see ‘Additional evidence for invariant symbolic representations’ below).
> - Symbol processing mechanisms employ *indirection*, meaning that variables refer to content that is stored at a different location than the variables themselves (i.e., they are pointers). In the identified architecture, the retrieval heads use the inferred symbols to retrieve the associated tokens from earlier positions in the sequence. That is, the symbol representations function as pointers that identify the address of the to-be-retrieved tokens.
>
> We will add an explicit statement to the revised paper that clearly defines symbol processing in these terms, and add more discussion explicitly relating this definition to the results.
>
> ## Additional evidence for invariant symbolic representations
> The RSA results showed that the outputs of abstraction and symbolic induction heads preserve token identity to some extent, which may seem to suggest that they do not represent abstract variables in an invariant manner. However, it is possible for these heads to invariantly represent abstract variables within a specific subspace, while also representing token identity. To test for this possibility, we performed an experiment in which a linear decoder was trained to predict the abstract variable (A or B) based on the outputs of these heads, and tested on its ability to *generalize out-of-distribution to problems involving completely novel tokens*. The decoder achieved nearly perfect (>98%) accuracy for both types of heads (Table 3). These results demonstrate that a subspace exists in which abstract variables are represented in an invariant manner, despite the fact that concrete tokens are also represented in other regions of the embedding space (note also that tokens are represented much more weakly, as shown in Table 1).
>
> ## Testing smaller models
> Unlike the larger models that we tested, none of the GPT-2 variants were able to reliably perform the task (Figure 1), and they did not show robust evidence for the presence of abstraction heads (Figure 1 and 6). These results were also not consistent between the two rule types (ABA vs. ABB, see Figures 5 and 7), again suggesting a lack of robustness. These results suggest that symbolic mechanisms may only emerge at certain scales (whether in terms of model or training data size). These results also strengthen our argument that abstract reasoning in language models depends on the presence of emergent symbolic mechanisms – language models that do not develop these mechanisms are not able to reliably solve abstract reasoning tasks.
>
> ## Responses to other comments
> - Implications for designing more robust AI systems: One implication of these results is that the identified mechanisms could potentially be built directly into the architecture of language models. This has been explored to some extent in architectures such as the Abstractor (which implements a mechanism similar to abstraction heads), but could be taken further by incorporating the other mechanisms that we identify.
> - Impact of training regime and architecture: We agree that it would be very interesting to investigate how aspects of the training regime and model architecture contribute to the emergence of symbolic mechanisms. These experiments would necessitate very extensive resources, as they would involve training language models from scratch, but we consider this an important direction for future work.
> We will add further discussion of these issues to the revised paper.

---

### Decision · Program_Chairs · 2025-05-01

**Decision:**

Accept (poster)

**Comment:**

This paper investigates the internal mechanisms that support abstract reasoning in LLMs, focusing on the open-source model Llama3-70B. It is argued that emergent reasoning in neural networks depends on the emergence of symbolic mechanisms. Evidence for this claim is presented in the form of a discovery of an emergent symbolic architecture in LLama3-70b that implements abstract reasoning via a series of three computations: abstraction, induction and retrieval. Experiments are carried out on algebraic rule induction tasks.

This paper received mostly positives reviews and after the rebuttal all reviewers are in favor of accepting. Reviewers agree the interpretability techniques deployed are appropriate, and the results are sound. There is also broad agreement that the results presented are of broader interest, especially because of the new heads/circuits that are identified. A general weakness noted among the reviewers was a lack of results for other models and tasks, which the authors addressed in the rebuttal. Other remaining questions, such as about clarifying what is meant with "symbolic processing" were mostly addressed in the author response and will hopefully make their way into the paper. The AC concurs with the recommendation to accept.